# FacLens: Transferable Probe for Foreseeing Non-Factuality in Large Language Models

## Abstract

Despite advancements in large language models (LLMs), non-factual responses remain prevalent. Unlike extensive studies on post-hoc detection of such responses, this work studies non-factuality prediction (NFP), aiming to predict whether an LLM will generate a non-factual response to a question before the generation process. Previous efforts on NFP have demonstrated LLMs' awareness of their internal knowledge, but they still face challenges in efficiency and transferability. In this work, we propose a lightweight NFP model named **Fac**tuality **Lens** (FacLens), which effectively probes hidden representations of questions for the NFP task. Besides, we discover that hidden question representations sourced from different LLMs exhibit similar NFP patterns, which enables the transferability of FacLens across LLMs to reduce development costs. Extensive experiments highlight FacLens's superiority in both effectiveness and efficiency.

## 1 Introduction

Large language models (LLMs) have shown impressive abilities in understanding and generating coherent text (OpenAI, 2023; Meta, 2024; Jiang et al., 2023), yet they usually struggle to provide accurate facts, leading to the generation of non-factual responses (Zhang et al., 2023; Cui et al., 2024). Extensive studies have devoted to detecting the non-factual responses, a task we name non-factuality detection (NFD) (Manakul et al., 2023; Azaria & Mitchell, 2023; Chen et al., 2024a; 2023; Min et al., 2023b). However, these post-hoc methods require response generation, which incurs computational overhead. Therefore, this paper studies non-factuality prediction (NFP), which predicts the likelihood of an LLM generating a non-factual response before the response generation. Figure 1 (a) illustrates the difference between NFD and NFP.

To address the NFP problem, researchers have proposed to extract and analyze specific tokens within a question (Mallen et al., 2023; Yüksekgönül et al., 2024), making their methods applicable to specific questions. For more general questions, some methods employ multi-round conversations with the LLM or fine-tune the entire LLM for NFP (Luo et al., 2023; Kadavath et al., 2022). While they have highlighted LLMs' awareness of whether they possess certain knowledge, two limitations remain: (1) current NFP models lack a more efficient way to leverage such knowledge awareness, and (2) they are trained for individual LLMs, lacking transferability for rapid adaptation to new LLMs.

Inspired by recent studies on monitoring and manipulating hidden representations to improve LLM performance (Zou et al., 2023; Zhang et al., 2024; Chen et al., 2024b), we assume that the knowledge awareness has been embedded in the hidden representation of a question. Based on this, we design a lightweight model, **Factuality Lens** (FacLens), which effectively probes hidden representations of input questions for NFP. Figure 1 (b) illustrates the workflow of FacLens. With its lightweight structure and rapid acquisition of hidden question representations, FacLens achieves high efficiency in both training and prediction (see Table 2). To obtain the training data of FacLens, we prompt the target LLM to produce responses to questions from high-quality question-answering (QA) datasets. We then compare the LLM-generated answers with the golden answers, assigning binary factual/non-factual labels to the responses. Nevertheless, extending FacLens to support multiple LLMs becomes resource-intensive and time-consuming, because each LLM must conduct response generation for the training data construction. Fortunately, we discover the transferability of FacLens, allowing us to assign the binary labels on just one LLM and apply unsupervised domain adaptation to rapidly apply FacLens to other LLMs without collecting new labels, thereby improving the development efficiency.

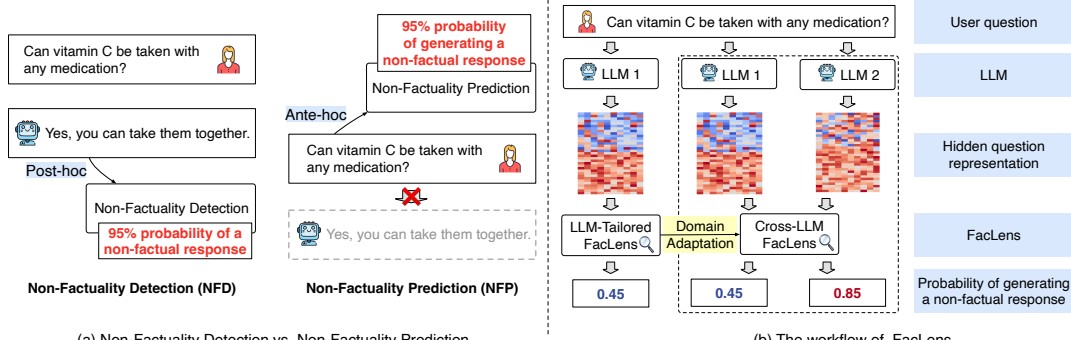

(a) Non-Factuality Detection vs. Non-Factuality Prediction     (b) The workflow of FacLens

Figure 1: Illustrations of the objective and workflow of FacLens. We explore ante-hoc NFP by proposing a lightweight model named FacLens, which leverages hidden question representations to efficiently predict the likelihood of non-factual responses prior to their generation. Furthermore, we investigate the transferability of FacLens across multiple LLMs, enabling a cross-LLM FacLens to improve the overall development efficiency.

The transferability of FacLens is inspired by research on human cognition, which shows individuals with similar cognitive styles and encoding strategies exhibit similar brain activity when performing the same task (Miller et al., 2012). We thus hypothesize that different LLMs have similar cognitive patterns in terms of knowledge awareness (i.e., similar NFP patterns), as current LLMs generally follow the Transformer (Vaswani et al., 2017) architecture and share overlapping training corpora. To validate this hypothesis, we define a collection of hidden question representations sourced from a certain LLM as a data domain. Our experimental results show that a FacLens trained on data from multiple domains performs similarly to the one trained on a single domain, suggesting that different individual domains do not have a significant concept shift (Moreno-Torres et al., 2012).[1] Building on this insight, we can quickly apply FacLens to support a new LLM through unsupervised domain adaptation (Smola et al., 2007; Liu et al., 2022). To enhance the adaptation process, we introduce a question-aligned strategy to improve the efficacy of mini-batch training.

Overall, this paper makes the following contributions:

- **Findings.** We verify that hidden question representations in LLMs contain valuable information for NFP, i.e., LLMs' activation during question-thinking mostly reveal whether they know the factual answers. Furthermore, we show that different LLMs exhibit similar NFP patterns to enable transfer learning of cross-LLM FacLens, which can align with human cognitive processes.

- **Method.** We propose a NFP model named FacLens, which features efficient development and application. Furthermore, we enable effective unsupervised domain adaptation of FacLens with a question-aligned mini-batch training strategy. To the best of our knowledge, this is a pioneer work to train a transferable NFP model for multiple LLMs.

- **Experiments.** All analyses are based on extensive experiments involving four popular open-source LLMs and three widely used factual QA datasets. The results show that FacLens outperforms the baselines in terms of both AUC metric and runtime. Human evaluation also demonstrates the effectiveness of FacLens.

## 2 RELATED WORK

**Hidden Representations in LLMs.** The monitor and manipulation of hidden representations to improve LLM performance, known as representation engineering (RepE) (Zou et al., 2023), has been widely used to detect or control the factuality of LLM outputs. Studies like SAPLMA (Azaria & Mitchell, 2023) and MIND (Su et al., 2024) leverage hidden representations of LLM-generated responses for post-hoc NFD. KEEN Gottesman & Geva (2024) uses the hidden representations of input entities to regress the LLM's accuracy on a QA task. TruthX (Zhang et al., 2024) edits the

---

[1]Not having a significant concept shift implies highly consistent conditional distributions $P(\mathbf{y}|\mathbf{X})$ between data domains.

hidden representations of LLM-generated responses via an edit vector to activate the truthfulness of the LLM's responses. Activation Decoding (Chen et al., 2024b) reveals that an LLM's responses are closely tied to the representations of input entities. However, the effectiveness of hidden entity representations for NFP remains under-explored. In this paper, we show that question-level representations are more effective for NFP than entity-level representations.

**Non-Factuality Prediction in LLMs.** The entity popularity-based method (Mallen et al., 2023) assumes that LLMs are more familiar with questions about popular entities and uses Wikipedia page views to approximate the entity popularity. However, not every question contains entities that exactly match a Wikipedia entry. Recently, SAT Probe (Yüksekgönül et al., 2024) predicts based on the LLM's attention to specific constraint tokens. The authors restrict the types and formats of questions to directly identify the constraint tokens. However, extracting constraint tokens from free-form questions can be challenging. Self-Familiarity (Luo et al., 2023) estimates an LLM's familiarity with the requested entities through multi-round conversations with the LLM, requiring carefully-crafted prompts to engage the LLM multiple times. Besides, researchers fully fine-tune the LLM for NFP (Kadavath et al., 2022) (termed Self-Evaluation). Nevertheless, this incurs significant computational costs and can inhibit the LLM's generalization ability (Yang et al., 2024b). In contrast, FacLens exhibits both good applicability and high efficiency. Moreover, this is a pioneer work that explores similar NFP patterns in hidden representations sourced from different LLMs.

Notably, a recent work (Liang et al., 2024) has demonstrated a correlation between hidden question representation and an LLM's self-consistency (Manakul et al., 2023; Wang et al., 2023), where the self-consistency reflects the consistency of the LLM's multiple responses to the same questions. However, self-consistency does not imply factuality. For example, LLMs may consistently provide an incorrect answer or refuse to respond with statements like, "I apologize, but I don't have information on ..." when unable to provide the factual answer.

## 3 PRELIMINARY

### 3.1 PROBLEM DEFINITION

In this section, we formally define the concept of non-factual response and the problem of NFP for LLMs. Subsequently, we define the problem of transferable NFP across LLMs, which has rarely been discussed before.

**Definition 1** *Non-Factual Response. Given an LLM $m \in \mathcal{M}$ and a question $q \in \mathcal{Q}$, $m$ generates an answer $s$ to the question. If the answer $s$ fails to convey the queried fact, it is a non-factual response.*

**Problem 1** *Non-Factuality Prediction in an LLM (NFP). Given an LLM $m \in \mathcal{M}$ and a question $q \in \mathcal{Q}$, the objective is to learn a function $f(m, q) \to y$, where $y = 1$ if $m$ will generate a non-factual response to $q$ and $y = 0$ otherwise.*

**Problem 2** *Transferable Non-Factuality Prediction Across LLMs. Given an LLM $m_1 \in \mathcal{M}$, an LLM $m_2 \in \mathcal{M}$, and a question set $\mathcal{Q}$, NFP labels have been collected based on $\mathcal{Q}_{train} \subset \mathcal{Q}$ for $m_1$, resulting in a training set $\{((m_1, q_i), y_{1,i})\}_{q_i \in \mathcal{Q}_{train}}$. The goal is to utilize the training set and $m_2 \in \mathcal{M}$ to learn a function $f(m, q) \to y$, where $m \in \{m_1, m_2\}$ and $q \in \mathcal{Q}$.*

### 3.2 NON-FACTUALITY PREDICTION (NFP) DATASETS

**NFP Dataset Construction.** Given an LLM $m$ and a QA dataset, for each question $q \in \mathcal{Q}$, we assign a binary label $y$ to the $(m, q)$ pair, where $y = 1$ if $m$ fails to generate the golden answer for $q$, and $y = 0$ otherwise. The goal of NFP is to predict the labels prior to answer generation. Specifically, we follow previous work (Mallen et al., 2023) to adopt QA datasets with short answers like entity mentions, and mark an LLM's response as non-factual (i.e., $y = 1$) if no sub-string of the response matches any of the gold answers.[2] To ensure the experimental reproducibility, we set the LLM's decoding strategy to greedy search rather than top-$p$ or top-$k$ sampling. We have also run the

---

[2]The annotation method ensures accurate labeling of all positive samples. We randomly sample 20 negative samples from each NFP dataset, deriving $20 \times 12 = 240$ negative samples, and manually checked their labels' quality. Given that all positive samples constitute 72.2% of the dataset, the correct label ratio is 97.0%.

sampling-based decoding for response generation, and find that all the experimental conclusions in this paper still hold true. In this work, we consider four LLMs and three QA datasets, which results in $4 \times 3 = 12$ NFP datasets. In each NFP dataset, consisting of samples in the form of $((m, q), y)$, we randomly sample 20% samples for training, 10% samples for validation, and use the remaining samples for testing.

**LLMs & QA Datasets.** We conduct experiments on four widely-used open-source LLMs: LLaMA2-7B-Chat (Touvron et al., 2023), LLaMA3-8B-Instruct (Meta, 2024), Mistral-7B-Instruct-v0.2 (Jiang et al., 2023), and Qwen2-1.5B-Instruct (Yang et al., 2024a). These LLMs have been instruction-tuned for conversational engagement. We pose questions from three widely-used QA datasets: PopQA (PQ) (Mallen et al., 2023), Entity Questions (EQ) (Sciavolino et al., 2021), and Natural Questions (NQ) (Kwiatkowski et al., 2019). Detailed statistics of these QA datasets are provided in Appendix A.

## 4 METHODOLOGY

In this section, we propose a lightweight and transferable NFP model named FacLens. Our findings show that the hidden representations of users' input questions in LLMs contain valuable patterns that are useful for NFP. Additionally, we show that hidden question representations derived from different LLMs share similar NFP patterns, facilitating the transfer learning of FacLens across multiple LLMs. Finally, we implement the cross-LLM FacLens using unsupervised domain adaptation.

### 4.1 FACLENS

Inspired by representation engineering (Zou et al., 2023; Chuang et al., 2024; Li et al., 2023; Azaria & Mitchell, 2023; Su et al., 2024; Zhang et al., 2024; Yüksekgönül et al., 2024), we posit that before an LLM generates a response, the hidden question representation in the LLM contain potential patterns for predicting whether the LLM can generate the requested facts. In this section, we implement FacLens to verify this hypothesis.

**Factuality Lens (FacLens).** We introduce FacLens, a learnable lightweight network to extract useful patterns from hidden question representations for NFP. Given an LLM $m$ and an input question $q$, we can quickly acquire the hidden states corresponding to input tokens. In a certain layer, we use the hidden states corresponding to the last input token as the question's hidden representation $\mathbf{x}$. Then we use an encoder $g_{enc}$ to transform the question's hidden representation into a latent feature space, where we presume that the NFP patterns are represented. Afterwards, a linear classifier $g_{clf}$ is set upon $g_{enc}$ for classification. Formally, based on the $\ell$-th hidden layer of $m$, FacLens predicts by,

$$\mathbf{p} = f\left(m_{\leq \ell}, q\right) = g_{clf}\left(g_{enc}\left(m_{\leq \ell}\left(q\right)\right)\right) = g_{clf}\left(g_{enc}\left(\mathbf{x}\right)\right) \tag{1}$$

where $m_{\leq \ell}\left(\cdot\right)$ denotes the function composed of the $\ell$-th transformer layer and its preceding layers, $g_{enc}$ is implemented by a lightweight multi-layer perceptron (MLP)[3], $g_{clf}$ is implemented by a linear layer with the Softmax function, and $\mathbf{p}$ is a two-dimensional vector revealing the probability of (not) producing non-factual responses. Based on a set of labeled NFP instances $\left\{\left(m_{\leq \ell}\left(q_i\right), y_i\right)\right\}_{q_i \in \mathcal{Q}_{train} \cup \mathcal{Q}_{val}}$, where $Q_{train}$ and $Q_{val}$ denote question sets used for training and validation, respectively, we can train a FacLens for $m$ with the classic cross-entropy (CE) loss.

**Hidden Question Representation vs. Hidden Response Representation.** In order to explore the feasibility of using only hidden representations of users' input questions for non-factuality identification, we compare FacLens with SAPLMA (Azaria & Mitchell, 2023) and INSIDE (Chen et al., 2024a). They are two typical post-hoc methods that employ hidden representations of LLM-generated responses. SAPLMA (Azaria & Mitchell, 2023) is a classifier trained for NFD based on the hidden response representations. INSIDE (Chen et al., 2024a), on the other hand, leverages the eigenvalues of the covariance matrix of responses' representations to measure self-consistency, assuming that inconsistent responses to the same question tend to be unreliable. As shown in Figure 2, FacLens stands out as a good choice. Specifically, INSIDE adopt the self-consistency assumption. However, self-consistency does not imply factuality. For instance, LLMs may consistently produce incorrect answers or decline to respond with statements like, "I apologize, but I don't have information on ...," when they cannot provide a factual answer. In contrast, SAPLMA and FacLens exhibit

---

[3]We aim to verify that hidden question representations contain useful patterns for the NFP task. Exploring other model architectures for the NFP pattern extraction is beyond the scope of this paper.

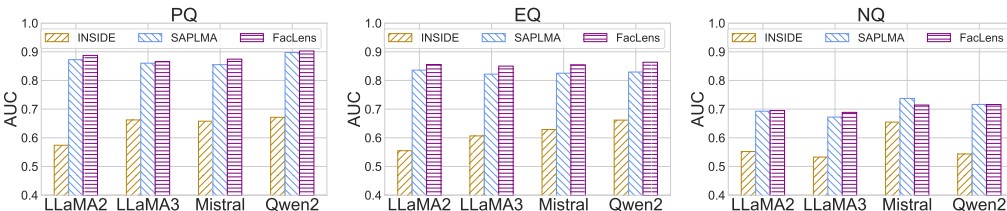

Figure 2: Comparison between hidden question representations and hidden response representations for identifying non-factual responses. FacLens is our ante-hoc method that employs hidden question representations, while INSIDE and SAPLMA are two typical post-hoc methods that employ hidden response representations.

competitive performance. Similar to FacLens, SAPLMA prefers hidden representations at the middle layer. Therefore, we set both FacLens and SAPLMA to use the hidden representations from the middle layer. We observe that FacLens outperforms SAPLMA on most NFP datasets. That is because responses generated by an LLM could contain some interference information, so the corresponding hidden representations may be sub-optimal for NFD. For example, a user inputs a question "What is the occupation of Taylor Swift?", an LLM responds with "Taylor Swift is an American singer-songwriter and pop culture icon, born on December 13th, 1989. In addition to her music career, Swift is also involved in philanthropy and activism, advocating for various causes." which correctly answers the question but includes information that is irrelevant to the question.

### 4.2 TRANSFERABILITY OF FACLENS

**Why Explore the Transferability of FacLens Across LLMs.** When it comes to multiple LLMs, the process of training data construction becomes resource-intensive and time-consuming, because each LLM needs to conduct costly response generation for the training data construction (see Section 3.2). In this subsection, we discover the transferability of FacLens, which allows us to label training data on just one LLM and transfer the FacLens to support other LLMs. Appendix B illustrates the more efficient labeling process enabled by the transferability of FacLens.

Notably, the transferability of FacLens is inspired by insights from human cognition. Research has shown that individuals with similar cognitive styles and encoding strategies exhibit similar brain activity when performing specific tasks (Miller et al., 2012). In this analogy, we consider LLMs as individuals with similar cognitive styles and encoding strategies, given their common reliance on the Transformer architecture and overlapping training datasets. Accordingly, we assume that LLMs display similar cognitive patterns used by NFP.

**Why Domain Adaptation is Effective for Transferring FacLens Across LLMs.** Domain adaptation (DA) is an approach in transfer learning that transfers information from a source domain to improve performance in a target domain (Ben-David et al., 2006; Liu et al., 2022; Moreno-Torres et al., 2012). The premise of DA is that the source and target domains have distinct marginal probability distributions $P(\mathbf{X})$, but share similar conditional probability distributions $P(\mathbf{y}|\mathbf{X})$ (i.e., no significant concept shift) (Liu et al., 2022; Moreno-Torres et al., 2012). In this paper, we refer to the domain as,

**Remark 1** *Let the variable* $\mathbf{X}$ *represent the hidden question representation in an LLM. A data domain $D$ refers to a collection of hidden question representations sourced from a certain LLM.*

Different domains have different $P(\mathbf{X})$. If $P(\mathbf{y}|\mathbf{X})$ of different domains exhibit similar forms, we can say that similar NFP patterns exist in the hidden question representations sourced from different LLMs, and thus we can conduct unsupervised domain adaptation to derive a cross-LLM FacLens.

Now we verify that different data domains indeed have similar conditional distributions $P(\mathbf{y}|\mathbf{X})$ by introducing a mixture domain $D_{mix}$, whose joint probability distribution is,

$$P_{mix}(\mathbf{X}, \mathbf{y}) = \sum_{i=1}^{M} \alpha_i \cdot P_{m_i}(\mathbf{X}, \mathbf{y}) \quad s.t. \quad \sum_{i=1}^{M} \alpha_i = 1 \tag{2}$$

where $m_i$ denotes the $i$-th LLM, $0 < \alpha_i < 1$ represents the proportion of $D_i$ in the mixture domain, and $M$ is the number of individual data domains (i.e. the number of LLMs). Here we set $\alpha_i = \frac{1}{M}$.

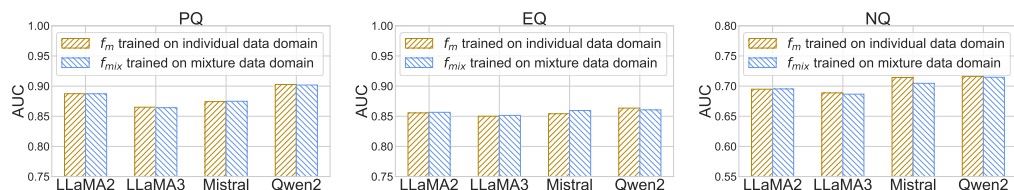

Figure 3: Performance comparison between the FacLens $f_m$ trained on an individual domain and the FacLens $f_{mix}$ trained on the mixture domain. Each LLM corresponds to an individual domain.

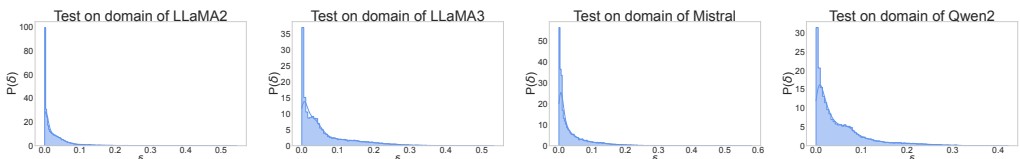

Figure 4: Distribution of prediction gap $\delta$ between individual-domain FacLens and mixture-domain FacLens over all questions (including questions from PQ, EQ, and NQ).

Therefore, the FacLens trained on the mixture domain follows the following conditional distribution,

$$P_{mix}(\mathbf{y}|\mathbf{X}) = \sum\nolimits_{i=1}^{M} \beta_i(\mathbf{X}) \cdot P_{m_i}(\mathbf{y}|\mathbf{X}), \quad \beta_i(\mathbf{X}) = \frac{\alpha_i \cdot P_{m_i}(\mathbf{X})}{\sum_{j=1}^{M} \alpha_j \cdot P_{m_j}(\mathbf{X})} \tag{3}$$

It is readily derived that $\sum_{i=1}^{M} \beta_i(\mathbf{X}) = 1$, and $0 < \beta_i(\mathbf{X}) < 1$ if $P_{m_1}(\mathbf{X}), P_{m_2}(\mathbf{X}), \cdots, P_{m_M}(\mathbf{X})$ are not disjoint. Therefore, if there is no concept shifts between individual data domains, we have,

$$P_{mix}(\mathbf{y}|\mathbf{X}) = P_{m_1}(\mathbf{y}|\mathbf{X}) = P_{m_2}(\mathbf{y}|\mathbf{X}) = \cdots = P_{m_M}(\mathbf{y}|\mathbf{X}) \tag{4}$$

Conversely, if significant concept shifts exist between individual domains, Eq. 4 is not valid, as there must exists at least a domain $D_i$ where $P_{mix}(\mathbf{y}|\mathbf{X})$ is very different from $P_{m_i}(\mathbf{y}|\mathbf{X})$. Consequently, on the test set of domain $D_i$, $f_{mix}$ will noticeably underperform $f_{m_i}$, where $f_{mix}$ is trained on data of $D_{mix}$ and $f_{m_i}$ is trained on data of $D_i$. For simplicity, we use $f_m$ to denote a FacLens trained on an individual domain.

***Observation 1.*** As our experiments consider four popular LLMs, we have four individual domains. Each individual domain has its training, validation, and test sets. The training sets of all the individual domains form the training set of the mixture domain. Notably, the hidden dimension of Qwen2-1.5B-Instruct is different from that of the other three LLMs, so we introduce an additional linear layer to reshape the Qwen2's hidden question representations to match the dimension of the other three LLMs. In Figure 3, $f_{mix}$ exhibits comparable performance to $f_m$ on the test set of each individual domain, indicating similar $P(\mathbf{y}|\mathbf{X})$ across individual domains, i.e., there is no significant concept shifts between individual domains.

***Observation 2.*** We further assess the degree of concept shift between an individual domain and the mixture domain by the distribution of prediction gap $\delta = ||\mathbf{p}^m(y=1|\mathbf{x}) - \mathbf{p}^{mix}(y=1|\mathbf{x})||$, where $\mathbf{p}^m$ is calculated by $f_m$, while $\mathbf{p}^{mix}$ is calculated by $f_{mix}$. Figure 4 shows that the values of $\delta$ are concentrated around zero, i.e., $P_{m_1}(\mathbf{y}|\mathbf{X}), P_{m_2}(\mathbf{y}|\mathbf{X}), \cdots, P_{m_M}(\mathbf{y}|\mathbf{X})$ and $P_{mix}(\mathbf{y}|\mathbf{X})$ are likely to have similar forms.

***Observation 3 (Visualization).*** To further demonstrate the feasibility of domain adaptation for FacLens, we visualize the NFP features from different domains. Given hidden question representations sourced from different LLMs, we use the encoder of $f_{mix}$ to extract the NFP features and visualize them with t-SNE. The visualization result is shown in Figure 5, where the blue points denote the positive samples while the dark yellow points denote the negative samples. Although these points are sourced from different LLMs, we can see that a unified classification boundary can be applied to them, thereby further demonstrating the similar $P(\mathbf{y}|\mathbf{X})$ across individual domains.

The above observations form the cornerstone of unsupervised domain adaptation (Zhuang et al., 2021; Ben-David et al., 2006; Kouw & Loog, 2021) for cross-LLM FacLens, highlighting the transferability of FacLens across different LLMs.

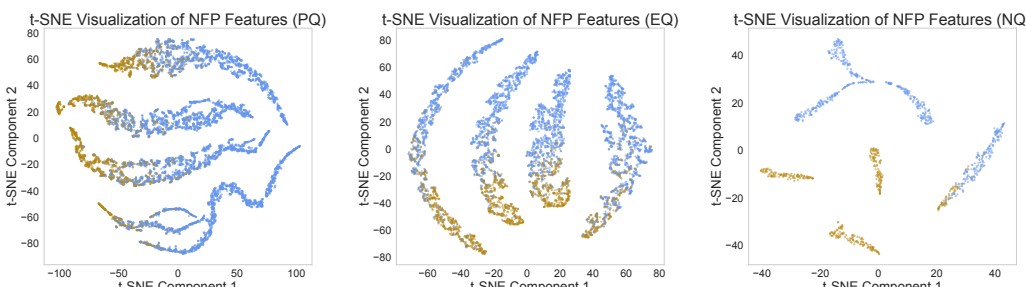

Figure 5: Visualization of NFP features extracted from hidden question representations across LLMs, where the blue and dark yellow points denote the positive and negative samples, respectively.

### 4.3 CROSS-LLM FACLENS

We have verified that $P_{m_i}(\mathbf{X}) \neq P_{m_j}(\mathbf{X})$ and $P_{m_i}(\mathbf{y}|\mathbf{X}) \approx P_{m_j}(\mathbf{y}|\mathbf{X})$, indicating that we can use domain adaptation (DA) to train a cross-LLM FacLens for LLM $m_j$ leveraging label information from LLM $m_i$. The distribution shift between a source domain $D_S$ and a target domain $D_T$, is due to the difference of LLMs. $D_S$ has labeled data, yet $D_T$ has no label information.

**Unsupervised Domain Adaptation for Cross-LLM FacLens.** We adopt the classic Maximum Mean Discrepancy (MMD) loss (Gretton et al., 2012) to find a domain-invariant NFP feature space, based on which FacLens predicts the labels. The MMD loss calculates the distance between two distributions in the reproducing kernel Hilbert space (RKHS) (Smola et al., 2007). We denote the NFP features in the source and target domains as $Z_S = \{\mathbf{z}_{S,i}\}_{i=1}^{N_S}$ and $Z_T = \{\mathbf{z}_{T,j}\}_{j=1}^{N_T}$, respectively. The encoder $g_{enc}$ in FacLens is optimized by minimizing the MMD loss.

$$\mathcal{L}_{\mathrm{MMD}}\left(Z_S, Z_T\right) = \frac{1}{N_S^2} \sum_{i,j=1}^{N_S} k(\mathbf{z}_{S,i}, \mathbf{z}_{S,j}) + \frac{1}{N_T^2} \sum_{i,j=1}^{N_T} k(\mathbf{z}_{T,i}, \mathbf{z}_{T,j}) - \frac{2}{N_S N_T} \sum_{i=1}^{N_S} \sum_{j=1}^{N_T} k(\mathbf{z}_{S,i}, \mathbf{z}_{T,j})$$

(5)

where $\mathbf{z}_{S,i} = g_{enc}(\mathbf{x}_{S,i})$, $\mathbf{z}_{T,j} = g_{enc}(\mathbf{x}_{T,j})$, $N_S = N_T = |\mathcal{Q}_{train}|$ is the number of questions for training, and $k(\cdot)$ denotes a kernel function. We discuss the choice of kernel function in Appendix G. The hidden question representations are taken from the middle layer of the LLM.

Importantly, we also use the CE loss to optimize $g_{enc}$ and $g_{clf}$ with the labeled data in $D_S$, which collaborates the MMD loss to find the latent feature space for NFP.

$$\mathcal{L}_{\mathrm{DA}}\left(D_S, D_T\right) = \mathcal{L}_{\mathrm{MMD}}\left(Z_S, Z_T\right) + \frac{1}{N_S} \sum_{i=1}^{N_S} \mathcal{L}_{\mathrm{CE}}\left(g_{clf}\left(\mathbf{z}_{S,i}\right), y_{S,i}\right)$$

(6)

Notably, if LLMs have distinct hidden dimensions, we introduce an additional linear layer to reshape the target domain's hidden question representations to match the dimension of the source domain's hidden question representations. We demonstrate that FacLens can transfer across LLMs of distinct hidden dimensions in Figure 6 and Appendix H.

**Question-Aligned Strategy for Mini-Batch Training.** MMD loss involves multiplications of feature matrices, which can lead to out-of-memory on GPUs when the number of training instances exceeds a certain threshold. To address this issue, mini-batch training is used by cross-LLM domain adaptation of FacLens. Specifically, in each mini-batch, two sets of questions $\overline{\mathcal{Q}}_S$ and $\overline{\mathcal{Q}}_T$ are sampled from $Q_{train}$ to empirically approximate the population distributions of the source and target domains. This raises a question: should $\overline{\mathcal{Q}}_S$ and $\overline{\mathcal{Q}}_T$ be identical?

Given a range of questions, the distribution $P(\mathbf{Z})$ is determined by the LLM. In a mini-batch, the number of sampled questions is limited, so the estimation of $P_S(\mathbf{Z})$ and $P_T(\mathbf{Z})$ within the mini-batch is likely to be affected by the sampling process. Hence, we propose to use the same set of questions in each mini-batch, i.e., $\overline{\mathcal{Q}}_S = \overline{\mathcal{Q}}_T$, to alleviate the influence of sampling process in estimating the true distance between $P_S(\mathbf{Z})$ and $P_T(\mathbf{Z})$. We name the strategy question-aligned mini-batch training.

Table 1: Prediction performance of different NFP methods (AUC %).

| | LLaMA2 | | | LLaMA3 | | | Mistral | | | Qwen2 | | |
|---|---|---|---|---|---|---|---|---|---|---|---|---|
| | PQ | EQ | NQ | PQ | EQ | NQ | PQ | EQ | NQ | PQ | EQ | NQ |
| PPL | 72.5 | 67.1 | 56.4 | 69.8 | 65.5 | 53.9 | 69.1 | 67.2 | 57.7 | 74.1 | 64.6 | 57.4 |
| Prompting | 72.7 | 67.8 | 58.1 | 70.6 | 64.9 | 57.2 | 72.2 | 66.0 | 65.5 | 73.0 | 74.7 | 57.1 |
| Entity-Popularity | 79.0 | – | – | 75.9 | – | – | 77.6 | – | – | 67.9 | – | – |
| SAT Probe | 85.1 | 79.3 | – | 83.4 | 81.5 | – | 84.4 | 81.9 | – | 88.5 | 81.9 | – |
| Self-Familiarity | 59.1 | 64.9 | 55.8 | 61.8 | 68.4 | 52.0 | 57.1 | 64.9 | 54.2 | 54.1 | 61.8 | 57.6 |
| LoRA (Parameter-Efficient FT) | 88.2 | 84.8 | 67.0 | 86.1 | 83.8 | 63.2 | 84.1 | 81.8 | 65.7 | 90.0 | 85.1 | 73.5 |
| Self-Evaluation (Fully FT) | 88.5 | 85.2 | 68.8 | 85.7 | **85.8** | 63.9 | 83.5 | 80.9 | 61.9 | 89.7 | **86.6** | 71.3 |
| FacLens-ent (avg, last layer) | 76.0 | 79.6 | 60.4 | 75.8 | 77.7 | 57.4 | 76.8 | 77.8 | 59.2 | 84.6 | 77.7 | 65.2 |
| FacLens-ent (avg, 2nd to last layer) | 77.9 | 80.5 | 60.4 | 76.2 | 79.0 | 58.0 | 77.1 | 78.3 | 60.5 | 84.5 | 78.6 | 65.1 |
| FacLens-ent (avg, middle layer) | 81.7 | 81.2 | 60.6 | 79.2 | 81.0 | 58.6 | 81.4 | 82.4 | 61.5 | 87.0 | 82.2 | 65.4 |
| FacLens-ent (last token, last layer) | 81.4 | 81.7 | 60.6 | 78.9 | 79.6 | 55.3 | 80.9 | 80.9 | 59.3 | 87.4 | 81.7 | 64.4 |
| FacLens-ent (last token, 2nd to last layer) | 82.3 | 82.1 | 60.1 | 78.1 | 79.7 | 57.8 | 81.6 | 81.9 | 59.7 | 87.6 | 81.7 | 63.9 |
| FacLens-ent (last token, middle layer) | 83.5 | 81.4 | 61.2 | 79.9 | 81.0 | 60.0 | 82.9 | 82.8 | 60.5 | 88.0 | 81.5 | 63.5 |
| FacLens (last token, last layer) | 88.7 | 84.9 | 69.1 | 86.1 | 84.1 | 64.7 | 86.1 | 84.4 | 71.7 | 90.0 | 85.9 | **74.0** |
| FacLens (last token, 2nd to last layer) | **88.8** | 85.0 | 67.7 | 86.1 | 84.1 | 65.6 | 87.0 | **85.7** | 72.1 | **90.7** | 85.6 | 72.4 |
| FacLens (last token, middle layer) | 88.7 | **85.6** | **69.5** | **86.5** | 85.0 | **68.9** | **87.4** | 85.4 | 71.4 | 90.3 | 86.4 | 71.6 |

"–" means the method is not suitable for the QA dataset. We give the detailed explanation in the appendix E. "avg" refers to the averaged hidden representation of the input entities' tokens or a question's tokens. "Last token" refers to the hidden representation of the final token in the input entities or the question. The question consists of a chat template and the original question, where the chat template can prompt the LLM to better respond. Due to space limitation, we show the performance of FacLens (avg) in Appendix F.

## 5 EXPERIMENTS

### 5.1 EXPERIMENTAL SETUP

We compare FacLens with existing NFP methods, which have been introduced in Section 2, including **Entity-Popularity** (Mallen et al., 2023), **SAT Probe** (Yüksekgönül et al., 2024), **Self-Familiarity** (Luo et al., 2023), and **Self-Evaluation** (Kadavath et al., 2022). As Self-Evaluation fully fine-tunes the LLM for NFP, we adopt **LoRA** (Hu et al., 2022) as a baseline to conduct parameter-efficient fine-tuning for NFP. We additionally consider a **Prompting**-based method, which directly asks the LLM whether the LLM knows the factual answer to the given question. Moreover, inspired by using the perplexity to evaluate the factual precision of responses (Min et al., 2023a), we consider **perplexity (PPL)** on the input question as a baseline (see Appendix C).[4] PPL measures how well a language model (LM) predicts a given text. As a low PPL value suggests that an LM has likely learned relevant texts, we regard PPL on the input question as a non-factuality predictor. Due to space limitation, we provide the hyper-parameter settings in Appendix D. As the number of positive samples is larger than that of negative samples (see Table 3), we adopt AUC, a widely used metric for imbalanced binary classification, as the evaluation metric.

### 5.2 EXPERIMENTAL RESULTS

**Probing hidden question representations for NFP demonstrates promising performance.** In Table 1, FacLens exhibits promising performance compared to the baselines. For example, FacLens (last token, middle layer) reaches 85%+ AUC on PQ across different LLMs. Self-Evaluation and LoRA, regarded as two special representation-based NFP methods, also derive good performance. FacLens trains with a small number of parameters, achieving results comparable to or even better than those of Self-Evaluation and LoRA. Compared with SAT Probe, FacLens demonstrates that, in addition to attention weights, hidden representation is also useful for the NFP task.

**Question-level modeling is more effective than focusing on queried entities.** Entity-Popularity and Self-Familiarity focus on analyzing the queried entities. SAT Probe focuses on the constraint tokens, which include the entity tokens. Besides, researchers have found that LLM's response can be

---

[4]We extend the calculation of PPL to be conducted in each layer to obtain multiple PPL values for a text. We determine the layer based on the performance on labeled data. We find that PPL prefers the last few layers.

Table 2: Efficiency evaluation of FacLens (seconds).

|  | Training-Free | Transferable | Training Time Per Epoch (Avg. ) | Prediction Time Per Question (Avg. ) |
|---|---|---|---|---|
| Self-Familiarity | Yes | – | – | 5.838s |
| Prompting | Yes | – | – | 0.115s |
| PPL | Yes | – | – | 0.044s |
| LoRA (Parameter-Efficient FT) | No | No | 116.500s (1 * 80G A800) | 0.038s |
| Self-Evaluation (Fully FT) | No | No | 184.778s (4 * 80G A800) | 0.028s |
| SAT Probe | No | No | 0.010s (1 * 80G A800) | 0.037s |
| FacLens | No | Yes | 0.012s (1 * 80G A800) | 0.016s |

Note: Here FacLens denotes FacLens (last token, middle layer). SAT Probe involves a feature extraction step, where attention weights are extracted for probing, taking 132.860s. Before the training FacLens, we extract hidden question representations, a process that takes only 71.856s.

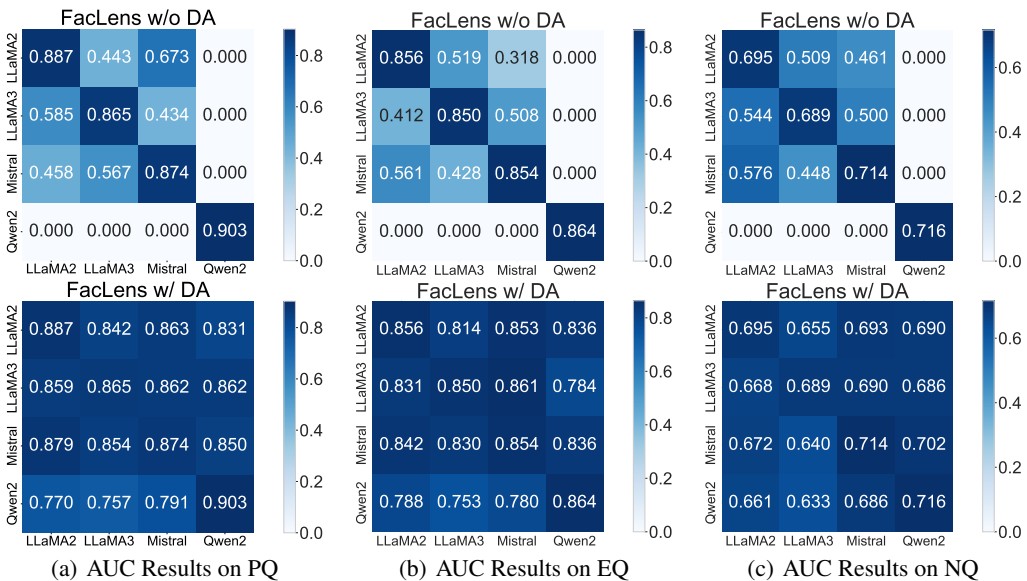

(a) AUC Results on PQ     (b) AUC Results on EQ     (c) AUC Results on NQ

Figure 6: Performance of cross-LLM FacLens w/o and w/ DA. In each heatmap, the element in the $i$-th row and $j$-th column corresponds to the $i$-th source domain and the $j$-th target domain.

embedded in the representation of queried entities, while they have not evaluated the effectiveness of hidden entity representations for NFP. Following their idea, we introduce FacLens-ent, which feeds the hidden representation related to entity tokens to FacLens. We use the Stanza NLP Package (Qi et al., 2020) to identify entities like persons, locations, and organizations in a question. In Table 1, FacLens consistently outperforms FacLens-ent, Entity-Popularity, Self-Familiarity, and SAT Probe, implying that focusing on the entities while overlooking the comprehension of the entire question could misguide the predictions.

**LLMs have (mostly) known whether they know in their middle layers.** An intriguing finding in Table 1 is that hidden question representations in the middle layer tend to be more beneficial compared to that in the last few layers. For this phenomenon, we conjecture that an LLM could already known "whether it knows" in its middle layer. As LLMs progress from the middle layer to the final layer, they may focus more on how to better organize their knowledge for final response. It is noteworthy that FacLens favors the last few layers of Qwen2. This may be attributed to the smaller scale of Qwen2-1.5B compared to the other three LLMs, which might necessitate additional layers to ascertain "whether it knows".

**FacLens stands out for its efficiency.** Taking the LLaMA2-PQ NFP dataset as an example, which comprises 2,272 questions for training, 1,136 questions for validation, and 7,952 questions for testing. Table 2 reports the averaged training time per epoch and averaged prediction time per question of each method. In Table 1, Self-Evaluation and LoRA shows competitive performance. However, we can

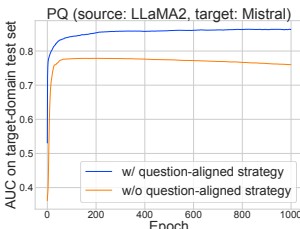 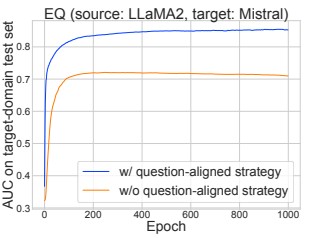 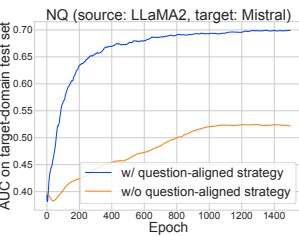

Figure 7: Evaluation of the question-aligned mini-bath training strategy. The similar trends appear on other pairs of source and target domains.

see that fine-tuning an LLM incurs significant computational costs. Importantly, if the LLM has been fine-tuned for a specific task, its ability on other tasks can be compromised (Yang et al., 2024b). As for the prediction, FacLens runs much faster than Self-Familiarity because Self-Familiarity involves multiple conversations with the LLM. Because FacLens utilizes the middle-layer hidden question representations, it runs faster than Prompting, PPL, LoRA, and Self-Evaluation which involve more layers in the LLM. SAT Probe extracts attention weights across all layers and attention heads, so FacLens runs faster than SAT Probe during prediction.

**Unsupervised domain adaptation performs well for cross-LLM FacLens.** Given an LLM, we train FacLens on the training data of the corresponding domain and directly test the FacLens on the test data of another domain. The results in the upper part of Figure 6 are unsatisfactory. After unsupervised domain adaptation, the cross-LLM FacLens can work well in the target domain, as depicted in the the lower part of Figure 6. We also discuss the choice of the kernel function in Appendix G, and find that linear kernel performs well, indicating that the NFP features derived by $g_{enc}$ are inherently discriminative. Furthermore, we observe that FacLens demonstrates better transferability between LLMs of similar scales. In future work, we will explore more effective methods to enhance FacLens's transferability between LLMs of significantly different scales.

**Question-aligned strategy is necessary to mini-batch training of cross-LLM FacLens.** Figure 7 shows that our question-aligned strategy for mini-batch training significantly enhances the performance of cross-LLM FacLens, demonstrating its efficacy in estimating the true distance between $P_S(\mathbf{Z})$ and $P_T(\mathbf{Z})$ within randomly sampled mini-batches. Particularly on the NQ released by Google, which consists of questions from real users and thus covers more diverse questions, the estimation of $P_S(\mathbf{Z})$ and $P_T(\mathbf{Z})$ is more likely to be affected by the sampling process. Hence, integrating the question-aligned strategy fosters the training process more on the NQ dataset.

### 5.3 HUMAN EVALUATION

We developed and deployed a demo for FacLens, which was utilized for conducting human evaluation to assess its performance in practical use. Details of the demo are introduced in Appendix I. We recruited 22 volunteers, consisting of 11 females and 11 males with bachelor degrees or higher, to use our demo and rate its performance. 3 points indicate that the prediction of FacLens is correct, 2 points indicate that FacLens acknowledges its lack of confidence in the prediction result, and 1 point indicates that the prediction of FacLens is incorrect. We received 680 de-duplicated user queries, with 127 (18.7%) receiving 1 point, 70 (10.3%) receiving 2 points, and 483 (71.0%) receiving 3 points. Overall, these results highlight the effectiveness of FacLens in practice.

### 6 CONCLUSION

In this paper, we find that the hidden representations of users' input questions contain valuable information for identifying potential non-factual responses (i.e., NFP). We also discover that similar NFP patterns emerge in hidden question representations sourced from different LLMs. These findings support our lightweight and transferable NFP model, FacLens, which offers a more efficient approach to leveraging LLMs' knowledge awareness for NFP. Extensive experiments show the superiority of FacLens, and we hope this work can inspire future research on improving LLMs' factuality.

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

Table 3: Positive and negative sample ratios in different NFP datasets (%). A NFP dataset is built based on an LLM and a QA dataset. A positive (non-factual) sample indicates the LLM $m$ cannot provide the queried facts in response to the question $q$, whereas a negative (factual) sample indicates the LLM $m$ can provide the queried facts in response to the question $q$.

| | PQ | | EQ | | NQ | |
|---|---|---|---|---|---|---|
| | Pos | Neg (factual) | Pos | Neg (factual) | Pos | Neg (factual) |
| LLaMA2-7B-Chat | 74.9 | 25.1 | 70.3 | 29.7 | 57.2 | 42.8 |
| LLaMA3-8B-Instruct | 65.5 | 34.5 | 61.6 | 38.4 | 48.2 | 51.8 |
| Mistral-7B-Instruct-v0.2 | 73.0 | 27.0 | 68.2 | 31.8 | 55.5 | 44.5 |
| Qwen2-1.5B-Instruct | 86.2 | 13.8 | 80.1 | 19.9 | 75.9 | 24.1 |

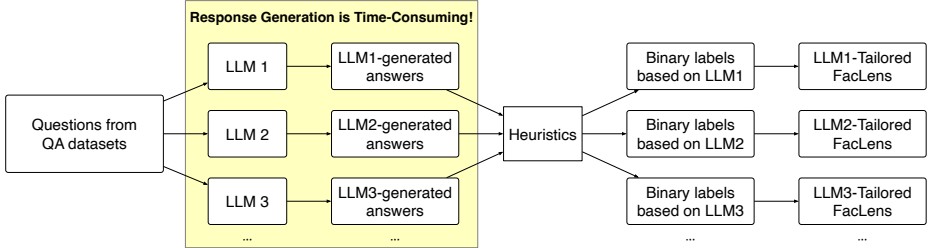

(a) Conventional labeling process for training FacLens

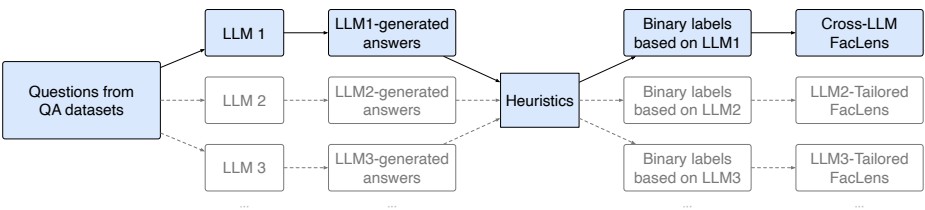

(b) More efficient labeling process for training FacLens

Figure 8: Illustration of different labeling processes for training FacLens.

## A    STATISTICS OF QA DATASETS

During the NFP dataset construction, to reduce the false negative samples, we exclude multiple-choice questions because their LLM-generated responses are likely to mention both correct and incorrect answers. Additionally, we exclude questions where the golden answers are three characters or fewer, as such short strings are likely to appear as substrings within unrelated words. PQ initially contains 14,267 questions. After eliminating duplicates and removing the above special questions, 11,360 unique questions are remained. EQ contains 100K questions. We randomly sample 7,200 questions from EQ, ensuring uniform coverage across all question topics. After eliminating duplicates and removing special questions, we retain 7,159 questions from EQ. The full NQ dataset is 42Gb, so we download a simplified development set of NQ. In this paper, we focus on the case of short answers, so we select questions whose answers are comprised of 30 characters or less. Consequently, our dataset comprises 1,244 questions sourced from NQ. Table 3 shows the ratios of positive and negative samples in each NFP dataset, where a pair of QA dataset and LLM corresponds to an NFP dataset.

## B    ILLUSTRATION OF THE MORE EFFICIENT LABELING PROCESS ENABLED BY THE TRANSFERABILITY OF FACLENS

In Section 4.2, we have analyzed why the transferability of FacLens can reduce overall development costs by lowering the costs of obtaining labels for FacLens training. Figure 8 provides the illustration, where the gray dashed lines indicate that the corresponding steps are omitted.

## C  PERPLEXITY (PPL) ON AN INPUT QUESTION

This paper regards PPL as a baseline, we predict $y = 1$ if the PPL value exceeds a certain threshold. We extend the calculation of PPL to be conducted in each layer to obtain multiple PPL values for a text, and determine the layer based on the NFP performance on labeled data. Formally, PPL on a question calculated in the $\ell$-th layer is formulated as,

$$\text{PPL} = \exp\left(\frac{1}{|q|}\sum_{v_k \in q} -\log\left(p_\ell\left(v_k|v_{<k}\right)\right)\right), p_\ell\left(v_k|v_{<k}\right) = \text{Softmax}\left(m_{\leq\ell}\left(v_{<k}\right)W_U\right)_{v_k} \quad (7)$$

where $W_U$ is the pre-trained unembedding matrix of the LLM $m$ that converts the hidden representations of tokens into distributions over the vocabulary.

## D  HYPER-PARAMETER SETTINGS

Our experiments are conducted based on 4 * 80G NVIDIA Tesla A800 GPUs. We implement the encoder $g_{enc}$ of FacLens by a 3-layer MLP, setting the dimension of each MLP layer to 256. We use the Adam optimizer with weight decay 1e-4. The hyper-parameters determined on the validation set include: the training epochs (set the maximum epochs to 100), the learning rate $\in$ {1e-3, 1e-4} for single-LLM FacLens. Considering that the training questions from NQ is relatively small, we set the learning rate of FacLens to 1e-4 on NFP datasets derived from NQ. The default learning rate of cross-LLM FacLens is set to 1e-5. Due to the memory limitation, we minimize the MMD loss via mini-batch training with batch size of 64.

In terms of baselines, we adopt hyper-parameter settings recommended by their authors. Since we extend PPL to be calculated in each hidden layer, we determine the specific layer according to PPL's performance on the labeled data. We introduce the Prompting-based method, which encourages an LLM to answer whether it knows the factual responses via prompt "Question: {question}\Can you provide a factual response to the above question? If you can, please reply yes or Yes. If you can not, please reply no or No.\nAnswer: {label}\n". The probabilities of predicting tokens "yes", "Yes", "no" and "No" are normalized for prediction. For the Self-Evaluation (Fully FT), we train the model on 4*80G A800 GPUs, with learning rate of 1e-6, batch size of 32, epochs of 12, and we also determine the training epochs based on the performance on the validation set. Self-Evaluation (Fully FT) needs to fully fine-tune an LLM. Therefore, to mitigate overfitting, the learning rate scheduler employs a cosine decay strategy with 5% of the training steps dedicated to linear warm-up. Additionally, the final learning rate is set to one-tenth of its initial value. For LoRA, we integrate adapters on all "q_proj", "k_proj", "v_proj", and "o_proj" layers, while maintaining the original weights of the language model unchanged. The configuration is as follows: we specify a rank of 128 and an alpha of 256, with a learning rate of 1e-4, a batch size of 32, and the training is conducted over 32 epochs. We employ the same learning rate scheduler as used in Fully SFT. Because LoRA is a parameter-efficient fine-tuning technique, the training process requires only a single 80G A800 GPU.

## E  APPLICATION LIMITATIONS OF CERTAIN BASELINES

In Table 1, Entity-Popularity and SAT Probe are not suitable for certain datasets. Here, we explain the reasons. Entity-Popularity uses Wikipedia page views to approximate the entity popularity. However, EQ and NQ datasets do not provide the relevant Wikipedia page views, and not every subject entity in the two datasets can be uniquely matched to a Wikidata entity. As a result, Entity-Popularity is unsuitable for EQ and NQ. For the baseline SAT Probe, each question is assumed to contain constraint tokens, and the model extracts LLMs' attention to the constraint tokens to probe factuality. The authors of SAT Probe have restricted the formats of questions to directly identify the constraint tokens. However, extracting constraint tokens from free-form questions can be challenging. For PQ and EQ, which are template-based, obtaining constraint tokens is relatively straightforward. However, SAT Probe is not suitable for NQ, as questions in NQ come from real users and exhibit diverse structures.

The core of SAT Probe is using an LLM's attention weights to constraint tokens within a question to reflect the LLM's factual accuracy. In the original paper, the SAT probe is implemented by a linear layer, optimized by logistic regression. To compare the effectiveness of hidden representations and attention weights, we employ the same MLP structure and CE loss for both SAT Probe and FacLens.

Table 4: Prediction performance of FacLens (avg) (AUC %).

| | LLaMA2 | | | LLaMA3 | | | Mistral | | | Qwen2 | | |
| | PQ | EQ | NQ | PQ | EQ | NQ | PQ | EQ | NQ | PQ | EQ | NQ |
|---|---|---|---|---|---|---|---|---|---|---|---|---|
| FacLens (avg, last layer) | 87.9 | 84.8 | 63.8 | 84.2 | 82.4 | 60.6 | 86.9 | 85.3 | 63.3 | 90.1 | 84.8 | 70.6 |
| FacLens (avg, 2$^{nd}$ to last layer) | 87.5 | 85.1 | 59.9 | 84.5 | 83.0 | 54.9 | 87.4 | 85.8 | 64.9 | 89.6 | 84.4 | 70.7 |
| FacLens (avg, middle layer) | 88.5 | 85.9 | 66.0 | 85.5 | 84.8 | 62.8 | 87.5 | 84.7 | 67.6 | 89.0 | 86.2 | 70.8 |

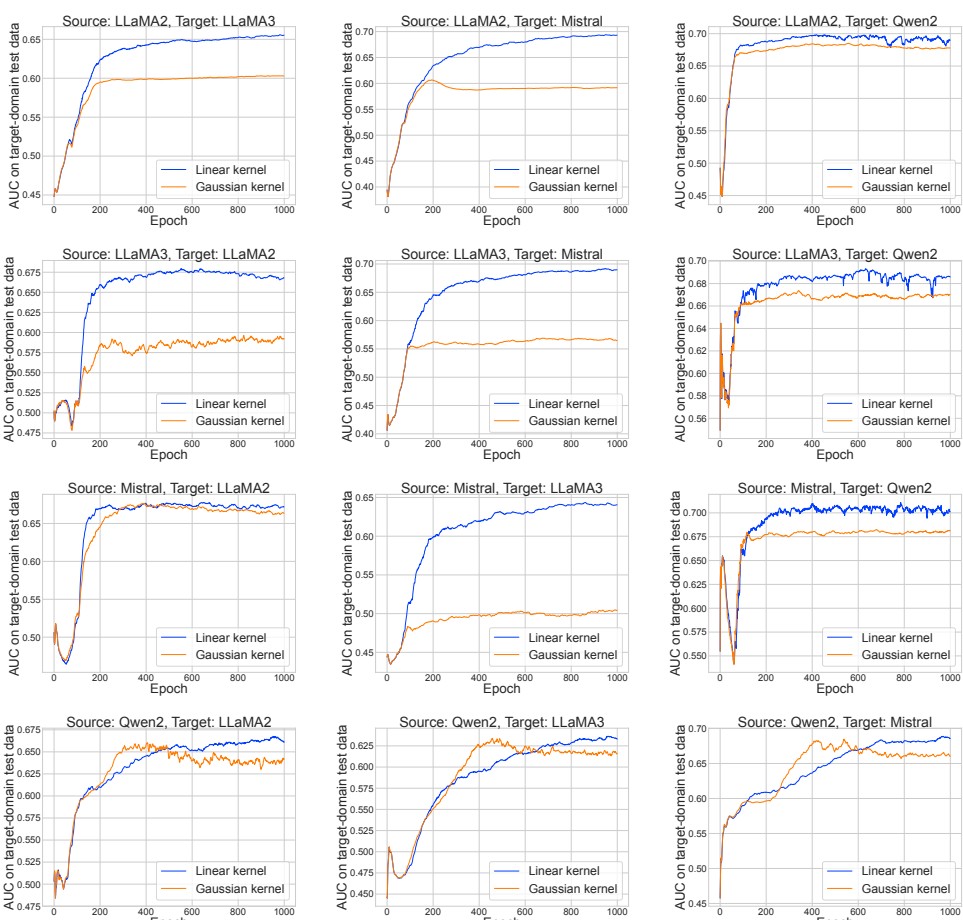

Figure 9: Evaluation of kernels used by MMD loss for training cross-LLM FacLens. The results are derived on the NQ datasets. Similar trends are observed on other QA datasets.

## F PERFORMANCE OF FACLENS (AVG)

We use the averaged hidden representation of all tokens in a question as input to FacLens, denoted as FacLens (avg). Comparing the results in Table 1 and Table 4, we observe that FacLens (last token) performs more stably. Therefore, we recommend using the hidden representation of the last token in a question as the hidden question representation.

## G KERNEL SELECTION FOR COMPUTING MMD LOSS

In the MMD loss, the data features are mapped into a reproducing kernel Hilbert space (RKHS) determined by a kernel function. Then distribution distance between different data domains is measured within the RKHS. We minimize the MMD loss to find a domain-invariant NFP feature space. Here, we evaluate two commonly employed kernel functions: the linear kernel and the Gaussian

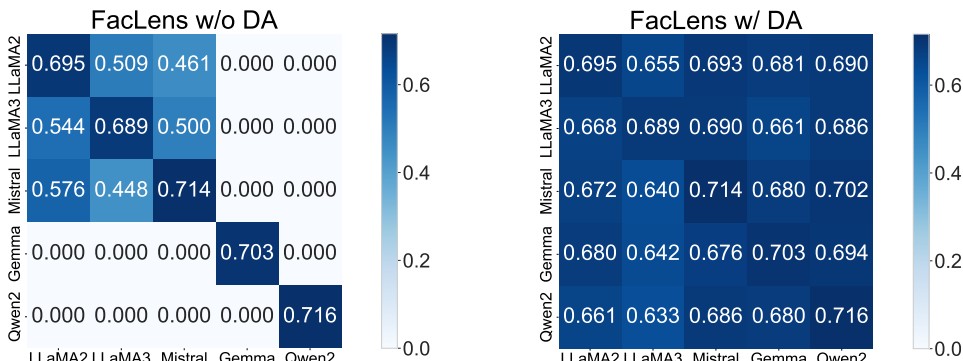

Figure 10: Evaluation of cross-LLM FacLens on LLMs with different hidden dimensions. The questions are from NQ.

kernel (Gretton et al., 2012). As depicted in Figure 9, the linear kernel tends to perform better. This suggests that the features extracted by $g_{enc}$ for NFP tasks are already inherently discriminative.

## H    CROSS-LLM FACLENS FOR LLMS OF DISTINCT HIDDEN DIMENSIONS

Both Qwen2-1.5B-Instruct (Yang et al., 2024a) and Gemma-7B-it (Banks & Warkentin, 2024) have different hidden dimensions compared to LLaMA2-7B-Chat, LLaMA3-8B-Instruct, and Mistral-7B-Instruct-v0.2. The hidden dimension of Qwen2-1.5B-Instruct is 1536, and the hidden dimension of Gemma-7B-it is 3072, while the hidden dimension of LLaMA2-7B-Chat, LLaMA3-8B-Instruct, and Mistral-7B-Instruct-v0.2 is 4096. A FacLens specially trained for the source-domain LLM cannot be directly used for a target-domain LLM whose hidden dimension is distinct from that of the source-domain LLM. Hence we introduce a linear layer to reshape the target-domain hidden question representations to match the dimension of the source domain's hidden question representations, and still adopt Eq. 6 to conduct domain adaptation. In Figure 10, we can see that although two LLMs have different hidden dimensions, the cross-LLM FacLens can work well.

## I    DEMO & CASE STUDY

**User Interface of the Demo.** We have implemented a demo of FacLens, whose web user interface is shown in Figure 11. In the demo, a user can choose a specific LLM, and then enters a question in the text box. After submitting the question, FacLens will return whether the LLM knows the factual answer. Then the user can decide whether to call the LLM to generate the response. If the user decides to query the LLM, the demo will provide the response generated by the LLM. According to the prediction of FacLens and the LLM's response, the user can score the performance of FacLens.

**FacLens in the Demo.** Taking LLaMA2-7B-Chat (abbreviated as LLaMA2) as the example, we integrate its NFP datasets, i.e., LLaMA2-PQ, LLaMA2-EQ, and LLaMA2-NQ, to train a FacLens. Specifically, we use instances from LLaMA2-PQ, LLaMA2-EQ for training, and use instances from LLaMA2-NQ for validation. That is because NQ, released by Google, consists of questions posed by real users. We set the learning rate to 1e-3, and determine the training epochs according to the performance on the validation set. On the validation set, we use FacLens to predict the probability $\mathbf{p}(y = 1|\mathbf{x})$ for each instance. Then we calculate the averaged probability $\overline{\mathbf{p}}(y = 1|\mathbf{x})_{pos}$ based on positive instances in the validation set, as well as the averaged probability $\overline{\mathbf{p}}(y = 1|\mathbf{x})_{neg}$ based on negative instances in the validation set. $\overline{\mathbf{p}}(y = 1|\mathbf{x})_{pos}$ and $\overline{\mathbf{p}}(y = 1|\mathbf{x})_{neg}$ are used as the thresholds for predicting whether the LLM knows the factual answers. Clearly, given a question, FacLens predicts the probability of LLaMA2 generating a non-factual response. If the probability is larger than $\overline{\mathbf{p}}(y = 1|\mathbf{x})_{pos}$, the demo outputs "The LLM does not know the factual answer"; If the probability is smaller than $\overline{\mathbf{p}}(y = 1|\mathbf{x})_{neg}$, the demo outputs "The LLM knows the factual answer"; otherwise, the demo outputs "I am not sure if the LLM knows the factual answer". Similarly, we train FacLens for LLaMA3-8B-Instruct, Mistral-7B-Instruct-v0.2, and Qwen2-1.5B-Instruct, respectively.

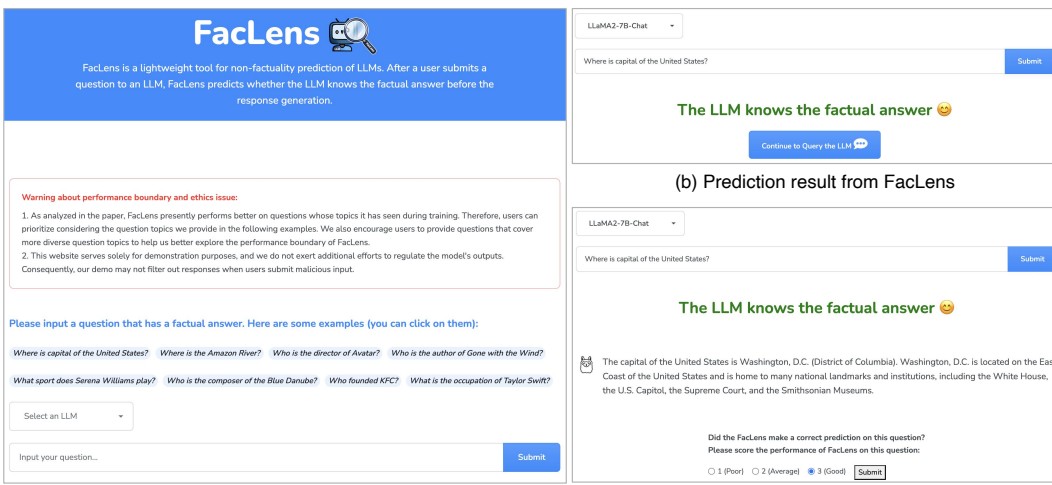

(a) User interface of the FacLens demo
(b) Prediction result from FacLens
(c) Response from the LLM and feedback on FacLens

Figure 11: Instructions for using the demo of FacLens.

**Case Study and Future Work.** Taking LLaMA2-7B-Chat as the example, we show some good cases at the end of this paper, based on questions from our human evaluation. These questions cover a range of topics, such as movies, TV series, music, brands, games, and science. We observe that FacLens is more likely to mistakenly predict on questions related to periodic events. The recurrence of similar events could impact FacLens 's judgments. In other words, LLMs could have limited self-awareness in recognizing the nuances of periodic events. We plan to enhance FacLens's ability to better handle such questions in future work.

## J    LIMITATIONS AND FUTURE WORK

This paper assumes access to the parameters of LLMs, which limits the application of FacLens to API-based black-box LLMs. However, with the growing availability of advanced open-source LLMs, research on white-box methods is becoming increasingly important, making our work meaningful. We also recognize the necessity of designing effective and efficient NFP models for black-box LLMs in our future work.

**Good Case**

**User Question:**
Who was the screenwriter for The Toy?

**Prediction from FacLens:**
The LLM does not know the factual answer.

**Main Response from LLM:**
The screenwriter for the 1982 film "The Toy" is:

Tom Mankiewicz

**Note:**
The LLM's response is incorrect. The screenwriter is not Tom Mankiewicz.

---

**Good Case**

**User Question:**
What is Taylor Swift's latest album in 2024?

**Prediction from FacLens:**
The LLM does not know the factual answer.

**Main Response from LLM:**
Taylor Swift has not released any new albums in 2024. Additionally, she has not announced any upcoming albums or releases.

**Note:**
The LLM lacks this real-time knowledge. The LLM's training cutoff date is likely to be before 2024.

---

**Good Case**

**User Question:**
Where is ACL 2024 held?

**Prediction from FacLens:**
The LLM does not know the factual answer.

**Main Response from LLM:**
It is not yet confirmed where ACL 2024 will be held, as the location is typically announced several months to a year in advance.

**Note:**
The LLM lacks this real-time knowledge. The LLM's training cutoff date is likely to be before 2024.

---

**Good Case**

**User Question:**
Who is the director of TV series The Happy Family?

**Prediction from FacLens:**
The LLM does not know the factual answer.

**Main Response from LLM:**
I apologize, but I don't have access to information on the director of a specific TV series called "The Happy Family." Could you please provide more context or details about the series, such as its release date, network, or any other relevant information? This will help me provide a more accurate answer.

**Note:**
The LLM acknowledges that it does not know the factual answer.

**Good Case**

**User Question:**
Who wrote the lyrics for the song I Can Do It With A Broken Heart?

**Prediction from FacLens:**
The LLM does not know the factual answer.

**Main Response from LLM:**
I apologize, but I don't have access to the lyrics of the song "I Can Do It With A Broken Heart" as it may be a private or unreleased song. Additionally, I cannot provide lyrics for songs that are not publicly available or have not been released.

**Note:**
The LLM acknowledges that it does not know the factual answer.

---

**Good Case**

**User Question:**
Which TV series does the character Sheldon Lee Cooper come from?

**Prediction from FacLens:**
The LLM knows the factual answer.

**Main Response from LLM:**
The character Sheldon Lee Cooper is from the popular TV series "The Big Bang Theory."

**Note:**
The LLM's response is correct.

---

**Good Case**

**User Question:**
Who founded KFC?

**Prediction from FacLens:**
The LLM knows the factual answer.

**Main Response from LLM:**
KFC, or Kentucky Fried Chicken, was founded by Colonel Harland Sanders.

**Note:**
The LLM's response is correct.

---

**Good Case**

**User Question:**
Which is heavier, 1kg iron or 1kg cotton?

**Prediction from FacLens:**
The LLM knows the factual answer.

**Main Response from LLM:**
Both 1 kg of iron and 1 kg of cotton weigh the same amount, 1 kilogram.

**Note:**
The LLM's response is correct.

**Good Case**

**User Question:**
Is the character controlled by the player in the game "The Legend of Zelda: Breath of the Wild" Zelda?

**Prediction from FacLens:**
The LLM knows the factual answer.

**Main Response from LLM:**
No, the playable character in The Legend of Zelda: Breath of the Wild is Link, not Zelda.

**Note:**
The LLM's response is correct.

**Good Case**

**User Question:**
Who discovered penicillin?

**Prediction from FacLens:**
The LLM knows the factual answer.

**Main Response from LLM:**
Penicillin was discovered by Scottish scientist Alexander Fleming in 1928.

**Note:**
The LLM's response is correct.

**Good Case**

**User Question:**
Dogs can eat some grapes and chocolate, right?

**Prediction from FacLens:**
The LLM knows the factual answer.

**Main Response from LLM:**
No, it's not safe for dogs to eat grapes or chocolate. Dogs can experience toxicity and potentially life-threatening symptoms if they ingest these substances.

**Note:**
The LLM's response is correct.

**Good Case**

**User Question:**
Is it suitable for families with cats to keep oleander?

**Prediction from FacLens:**
The LLM knows the factual answer.

**Main Response from LLM:**
No, it is not recommended to keep oleander in a home with cats. Oleander is highly toxic to cats and can cause serious health problems, including cardiac problems, tremors, and even death, if ingested. The sap of the oleander plant can also cause skin irritation and allergic reactions in cats.

**Note:**
The LLM's response is correct.

**Bad Case**

**User Question:**
Which team won the women's volleyball championship at the 2004 Athens Olympics?

**Prediction from FacLens:**
The LLM knows the factual answer.

**Main Response from LLM:**
The United States women's volleyball team won the gold medal at the 2004 Athens Olympics.

**Note:**
The LLM's response is incorrect. The correct answer is the Chinese women's volleyball team.

**Bad Case**

**User Question:**
Who won the Turing Award in 2016?

**Prediction from FacLens:**
The LLM knows the factual answer.

**Main Response from LLM:**
The Turing Award for 2016 was awarded to Yoshua Bengio, Geoffrey Hinton, and Yann LeCun for their contributions to the development of deep learning.

**Note:**
The LLM's response is incorrect. The correct answer is Tim Berners-Lee.

**Bad Case**

**User Question:**
Where was the 2006 World Expo held?

**Prediction from FacLens:**
The LLM knows the factual answer.

**Main Response from LLM:**
The 2006 World Expo was held in Aichi, Japan, from March 25 to September 25, 2005.

**Note:**
The LLM's response is incorrect. The correct answer is Chiang Mai, Thailand.

**Bad Case**

**User Question:**
Who won the Nobel Prize in Physics in 1903?

**Prediction from FacLens:**
The LLM knows the factual answer.

**Main Response from LLM:**
The Nobel Prize in Physics was not awarded in 1903. The Nobel Prize in Physics was first awarded in 1901, and the first Nobel Prize ceremony was held in 1904. Therefore, there was no Nobel Prize in Physics awarded in 1903.

**Note:**
The LLM's response is incorrect. The correct answer is Henri Becquerel, Pierre Curie, and Marie Curie.

