# OpenReview forum: "FacLens: Transferable Probe for Foreseeing Non-Factuality in Large Language Models"
_ICLR.cc/2025/Conference — Submitted to ICLR 2025_

### Official Review · Reviewer_aR8Q · 2024-11-03

**Soundness:** 3
**Presentation:** 3
**Contribution:** 2
**Rating:** 5
**Confidence:** 4

**Summary:**

This paper introduces FactLens, a probing method designed to predict whether a large language model (LLM) is likely to provide factual responses to a given question. Additionally, the authors demonstrate that FactLens can be effectively transferred across different models.

**Strengths:**

1. The method is clear and straightforward. FactLens is a simple probing method to assess question factuality, and its streamlined structure makes it easy to adopt.
2. Excellent efficiency and transferability. The experiments demonstrate that FactLens can be effectively transferred to other models, performing well across various benchmarks, including PopQA, Entity Questions, and Natural Questions.

**Weaknesses:**

1. The primary weakness is that FactLens does not show a clear performance improvement over previous methods. Both Figure 3 and Table 1 indicate that FactLens performs comparably to, but not significantly better than, prior approaches.
2. The experiment lacks a wider range of benchmarks. Adding more datasets, such as TriviaQA [1] and HotpotQA [2], could provide a more comprehensive evaluation.

[1] TriviaQA: A large-scale distantly supervised challenge dataset for reading comprehension. Joshi et al., 2017.
[2] HotpotQA: A dataset for diverse, explainable multi-hop question answering. Yang et al., 2018.

**Questions:**

Refer to the weakness.

---

> ### Author Response · Authors · 2024-11-18
> **Response to Reviewer aR8Q**
>
> Dear Reviewer aR8Q,
>
> We sincerely appreciate your effort in reviewing our paper. We are glad that you found it clear and recognized its excellent efficiency and transferability. Here we will carefully address your concerns and questions in detail.
>
> > **1. Performance Improvement**
>
> ***- Clarification on Figure 3:*** There may have been a misunderstanding regarding the meaning conveyed by Figure 3. The similar performance of $f_m$ and $f_{mix}$ is, in fact, a **positive outcome**. Both $f_m$ and $f_{mix}$ represent our FacLens model, differing only in training data. Our goal in comparing $f_m$ and $f_{mix}$ is to demonstrate FacLens’s transferability across LLMs. The similarity in their performance supports FacLens’s transferability across LLMs.
>
> ***- Clarification on Table 1:*** Table 1 shows that FacLens achieves clear performance gains over most baselines. While the performance gains over LoRA and Self-Evaluation are slightly smaller, FacLens significantly outperforms both of them in terms of training **efficiency** (see Table 2), which is crucial for practical applications. Moreover, as shown in Figure 2, we compared FacLens with post-hoc methods. Despite post-hoc methods having access to additional information (i.e., the generated answers), FacLens still performs better.
>
> > **2. Experimental Datasets**
>
> We adopted widely recognized datasets frequently used in the factual QA research, including the Natural Questions dataset, which is sourced from real user queries. Furthermore, we conducted human evaluation to collect real queries to further provide a practical assessment.
>
> Thank you very much for the valuable suggestion on incorporating more QA datasets! We supplemented main experiments with TriviaQA and HotpotQA, which are more complex QA datasets involving reasoning. The results demonstrate that our FacLens continues to perform well, and the domain adaptation is still effective.
>
> **Comparison between FacLens and NFP baselines on TriviaQA and HotpotQA**
> | Method                               | TriviaQA (LLaMA2) | HotpotQA (LLaMA2) | TriviaQA (LLaMA3) | HotpotQA (LLaMA3) |TriviaQA (Mistral) | HotpotQA (Mistral) |TriviaQA (Qwen2) | HotpotQA (Qwen2) |
> |--------------------------------------|-------------------|-------------------|-------------------|-------------------|-------------------|-------------------|-------------------|-------------------|
> | PPL                                  |58.4 | 55.2 | 54.6 | 55.2 |54.9 |54.4 |52.7 |53.5|
> | Prompting                            | 65.9 | 62.5 | 69.3 | 61.0 |73.8|63.0 | 57.6 | 62.1|
> | Entity-Popularity                  | - | - | - | - | - | - | - | - |
> | SAT Probe                            | - | - | - | - | - | - | - | - |
> | Self-Familiarity                     | 61.9 | 55.3 | 57.3 | 56.8 |59.3 | 54.5 |62.1 | 53.7 |
> | LoRA (Parameter-Efficient FT)                                | 71.7 | 72.9 | 51.6 | 68.0| 64.1 | 66.9 | 59.4 | 70.8 |
> | Self-Evaluation (Fully FT)                     |70.8 | 75.0 | 63.3 | 69.3 | 62.0 | 71.1 | 62.4 | 72.1 |
> | FacLens (last token, middle layer)     |      74.2         |   75.5            |      67.6         |      66.9      |69.2 | 74.9 | 70.3 | 71.4|
> | FacLens (last token, last layer) |      75.0       |       74.3        |    65.1           |        68.6       | 67.7 | 74.1 | 70.0 | 72.1 |
> | FacLens (last token, 2nd to last layer)   |      74.2         |          74.7     |     69.0          |      68.3        | 70.9 | 74.7 | 71.3 | 72.7|
>
> *Note: "-" indicates that the method is not applicable to the dataset. Compared to the most competitive baselines (i.e., LoRA and Self-Evaluation), FacLens demonstrates a significant improvement in training efficiency (see Table 2 in our paper).*
>
> **Evaluation of domain adaptation for cross-LLM FacLens on TriviaQA and HotpotQA**
> | Method              | w/o DA | w/ DA |
> |---------------------|-------------------|------------------|
> | LLaMA2 -> LLaMA3   (TriviaQA) |      44.7         |      64.5      |
> | LLaMA3 -> LLaMA2   (TriviaQA) |       57.3        |       70.5       |
> | LLaMA2 -> LLaMA3   (HotpotQA) |      46.2         |     67.3         |
> | LLaMA3 -> LLaMA2   (HotpotQA) |      38.7        |       70.9       |
>
> Through supplementing the suggested experiments, we found **another interesting observation**. We observed that for the more complex factual QA tasks (TriviaQA and HotpotQA), FacLens prefers hidden question representations from the later layers. This may be because LLMs tend to engage in deeper thinking when facing complex factual questions. We will incorporate this interesting observation to the next version of our paper.
>
> ---
>
> We sincerely thank you again for the constructive comments and the recognition of the strengths of our work. We would be delighted to engage in further discussion and address any additional questions you may have.
>
> Sincerely,
>
> All authors

---

> ### Author Response · Authors · 2024-11-25
> **Kind Request for Feedback**
>
> Dear Reviewer aR8Q,
>
> Thank you very much for taking the time to review our paper. We sincerely appreciate your valuable comments and have provided detailed responses to address your concerns.
>
> We would greatly appreciate it if you could let us know whether our responses have addressed your concerns. If there are any remaining questions, we would be more than happy to provide further clarification and work towards resolving them.
>
> Thank you once again for your time, effort, and consideration, and we look forward to your valuable feedback.
>
> Best regards,
>
> All Authors

---

> ### Author Response · Authors · 2024-11-25
> **A Kind Reminder to Check Our Responses**
>
> Dear Reviewer aR8Q,
>
> With the discussion period nearing its end, we notice that we have not yet received your feedback on our responses.
>
> We tried our best to address your concerns, providing detailed explanations and additional experimental results to clarify your concerns. We kindly request you to check our responses and reconsider your assessment of our paper.
>
> Thank you once again for your time and effort, and we look forward to hearing from you.
>
> Best regards,
>
> All authors

---

### Official Review · Reviewer_GNy3 · 2024-11-04

**Soundness:** 4
**Presentation:** 3
**Contribution:** 3
**Rating:** 8
**Confidence:** 4

**Summary:**

This work introduces FactLens, a NFP model which, unlike previous NFP methods, exhibits transferability across different LLMs. The authors make the following major contributions through the introduction of FactLens:
1. Show clear evidence of the importance of hidden layers in the pre-hoc/NFP setting.
2. Introduce a novel architecture for Factlens, such that the Factlens model weights can be adapted to a new LLM for good performance on the NFP task.
3. Conduct experiments to show superior performance in comparison to both post-hoc models, as well as similar NFP models. Factlens is also considerably faster than other similar models.

**Strengths:**

* This paper solves a series of important problems, notably the transferability in the NFP domain. This kind of domain adaptation is an important area of NLP research and should be encouraged across the community.
* The experiments are very well thought, extremely detailed, and the paper is overall pretty decently written.

**Weaknesses:**

Certain questions that seem to remain open:
* The size of LLMs used for training seems small. While this isn’t a major concern, it would be good to understand how FactLens does with larger models (say size > 50B)
* It’s not clear whether Domain Adaptation is used for the results in Table 1. If no, how does the domain-adapted Factlens do in comparison to other NFP baseline?. In general, the authors should clarify which of the results in the paper use DA for FactLens. This can be added in the experimental setup.

**Questions:**

Nothing major, see weaknesses.

---

> ### Author Response · Authors · 2024-11-18
> **Response to Reviewer GNy3**
>
> Dear Reviewer GNy3,
>
> We really appreciate your recognition of our paper as novel, important, and well-written. We are delighted that you consider our work an important contribution to the field of NLP and encourage its broader dissemination within the community.
>
> Your constructive suggestions will help us improve this paper. Here we will carefully address your questions in detail.
>
> > **1. Potential Use of LLMs with Larger Scales**
>
> Thank you very much for this suggestion! We acknowledge the potential of using LLMs with larger scales (e.g., over 50B) and plan to explore this direction as the future work. In practice, for many practitioners, training and deploying much larger LLMs presents substantial resource challenges, so they may still opt for LLMs under 10B. Our current work offers practical insights for them.
>
> > **2. Experimental Setup for the Domain Adaptation (DA)**
>
> Table 1 does not include the DA results. The reason is that existing baselines do not investigate and implement DA for cross-LLM Non-Factuality Prediction (NFP).
>
> In the DA setting, we have no labeled data for the target LLM. For a fair comparison, we compare the domain-adapted FacLens with unsupervised NFP baselines, including PPL, Prompting, Entity-Popularity, and Self-Familiarity. We summarize the results in the following table, where the domain-adapted FacLens demonstrates promising performance compared to the unsupervised NFP baselines. We appreciate your suggestion and will rephrase the related experimental setup.
>
> **Comparison Between Domain-Adapted FacLens with Unsupervised NFP Baselines on LLaMA3 (the same trend can be found on other LLMs)**
>
> | Method                                  | PQ  | EQ  | NQ |
> |-----------------------------------------|-------------|-------------|-------------|
> | PPL                                     | 69.8        | 65.5        | 53.9        |
> | Prompting                               |70.6        | 64.9        | 57.2        |
> | Entity-Popularity                       | 75.9        | -           | -           |
> | Self-Familiarity                        |61.8        | 68.4        | 52.0        |
> | Domain-Adapted FacLens (Qwen2 -> LLaMA3) | 75.7 | 75.3 | 63.3|
> | Domain-Adapted FacLens (LLaMA2 -> LLaMA3) | 84.2 | 81.4 | 65.5 |
> | Domain-Adapted FacLens (Mistral -> LLaMA3) | 85.4 | 83.0 | 64.0 |
> |FacLens (w/o DA, trained with labeled data) | 86.5 | 85.0 | 68.9 |
>
>
> ---
>
> We sincerely thank you again for your thoughtful review and positive feedback. Your recognition is a great encouragement to us. We would be happy to engage in further discussion and address any additional questions you may have.
>
> Sincerely,
>
> All authors

---

> ### Author Response · Authors · 2024-11-25
> **Kind Request for Feedback**
>
> Dear Reviewer GNy3,
>
> Thank you very much for taking the time to review our paper. We sincerely appreciate your valuable comments and have provided detailed responses to address your concerns.
>
> We would greatly appreciate it if you could let us know whether our responses have addressed your concerns. If there are any remaining questions, we would be more than happy to provide further clarification and work towards resolving them.
>
> Thank you once again for your time, effort, and consideration, and we look forward to your valuable feedback.
>
> Best regards,
>
> All Authors

---

### Official Review · Reviewer_yYWm · 2024-11-04

**Soundness:** 2
**Presentation:** 3
**Contribution:** 3
**Rating:** 5
**Confidence:** 3

**Summary:**

Unlike studies on post-hoc detection of non-factual responses, this paper studied non-factuality prediction (NFP), that aims to predict whether an LLM will generate a non-factual response to a question before the generation process. While previous efforts on NFP have demonstrated LLMs' awareness of their internal knowledge, they still faced challenges in efficiency and transferability. Thus, this paper proposed a lightweight NFP model, named Factuality Lens (FacLens), which effectively probes hidden representations of questions for the NFP task. Further, this paper discovered that the hidden question representations from different LLMs exhibit similar NFP patterns, which enables the transferability of FacLens across LLMs to reduce development costs. The experiments highlighted FacLens’s superiority in both effectiveness and efficiency.

**Strengths:**

This paper tackled an interesting task of non-factuality prediction (NFP), tried to solve the problems of previous work in efficiency and transferability, and proposed a lightweight NFP model, named Factuality Lens (FacLens). The experiments highlighted FacLens’s superiority in both effectiveness and efficiency.

**Weaknesses:**

1. While the observations that the hidden question representations from different LLMs exhibited similar NFP patterns in Sec. 4.2 is interesting, we are eager to know why they happened and why unsupervised domain adaptation is possible in cross-LLM FacLens. It is better to investigate and mention some possible reasons for them, if possible, while the inspiration from the research on human cognition was mentioned in the paper.

Further, I wonder the observations in Sec. 4.2 can be really applicable to other LLMs. Can the authors mention the generalizability of the observations?

More seriously, in Sec. 5, depending on the datasets, the characteristics and the performance of LLMs seem different in Fig. 6. For example, on PQ and EQ, Qwen2 is rather different from the others, that leads to a concern that the assumption is not really correct and the transferability cannot be really valid among the LLMs.

2. I have a concern about the method for NFP dataset construction. Following previous work, the authors adopted a method for QA datasets with short answers. However, all current QA datasets are not generally in the category. It is better to show how large the method can cover the current QA datasets and/or to describe how they can cope with QA datasets with longer or more complex answers.

3. When the authors construct training data on just an LLM, the selection of the LLM might be important and might affect the performance. So it is better to show which LLM the authors selected as the best for constructing the training data and how they decided it.

4. In human evaluations in Sec. 5.3, it is better to have comparisons with some baselines.

**Questions:**

1. While the authors criticized the output statement in cases when LLMs cannot provide factual answers in the end of Sec. 2, I could not understand the criticization because the statement is not necessarily a non-factual answer. I hope the authors will clarify the point.

---

> ### Author Response · Authors · 2024-11-18
> **Response to Reviewer yYWm**
>
> Dear reviewer yYWm,
>
> We sincerely appreciate your constructive review and are delighted that you found our work interesting. Below, we will provide detailed responses to the questions raised.
>
> > **1. Explanation About the Tansferability of FacLens**
>
> ***- How to Explain Observations in Section 4.2***
>
> Current LLMs commonly share the similar architecture (i.e., Transformer), training scheme (i.e., pre-training, SFT, and RLHF), and overlapping knowledge resource (i.e., training corpora), e.g., LLMs such as Qwen and LLaMA, use a mix of data from publicly available sources. This suggests that LLMs are likely to have similar cognition styles.
>
> In human cognition, individuals with similar cognitive styles show similar brain activity when performing the same task. This inspires us to explain why hidden question representations from different LLMs (i.e., LLMs' brain activation during question-thinking) exhibit similar NFP patterns.
>
> ***- Why Unsupervised Domain Adaptation (DA) is Effective for Cross-LLM FacLens***
>
> The reason was explained in Section 4.2. In specific, we outlined the premise of unsupervised DA. Then we designed experiments (see Figures 3, 4, and 5) to demonstrate that cross-LLM FacLens satisfies this premise.
>
> ***- Generalization of the Observations in Section 4.2***
>
> In Section 4.2, we conducted experiments on various LLMs, including LLaMA2, LLaMA3, Mistral, and Qwen2. The results across different LLMs lead to the same observation (see Figures 3, 4, and 5), demonstrating the generalizability of our observations. For example, in each subfigure of Figure 5, samples from different LLMs form clusters and share the same classification boundary for NFP.
>
> ***- Correctness of the Transferability***
>
> The transferability of FacLens is validated by comparing results without and with DA, as shown in the upper and lower of Figure 6. Without DA (upper), FacLens trained on one LLM performs poorly on another, especially Qwen2, which differs from the other three LLMs in hidden dimensions and scale. With DA (lower), the performance of FacLens on the target LLM, including Qwen2, improves significantly.
>
> Note that Qwen2 used in this paper differs in scale from the other LLMs. As mentioned on page 10, while FacLens shows transferability between any two LLMs, better transferability is observed among LLMs of similar scales. This does not affect the validity of the demonstrated transferability. Future work will explore methods to enhance FacLens’s transferability across LLMs of different scales.
>
> > **2. QA Datasets with Long Answers**
>
> The factual QA scenario naturally favors short answers, as it focuses on querying specific real-world facts, such as names, numbers, birthplaces, dates, occupations, and so on. Additionally, most mainstream factual QA datasets adopt short answers. For these reasons, like previous works [1, 2, 3], this paper focuses on common factual QA, which typically involves short answers.
>
> While general QA with long-form answers (e.g., text summarization or literary creation) may include factual errors in generated responses, addressing factual errors in this context is another problem requiring a separate methodology, which presents a promising direction for future research.
>
> > **3. Selection of LLM for Constructing Labeled Training Data**
>
> The choice of LLM for constructing labeled training data is flexible. Our goal is to explore whether a FacLens trained for one LLM can quickly adapt to another. The unsupervised DA of cross-LLM FacLens has shown to be effective across any LLM pair (see Figure 6). When a specific LLM can be selected, based on our experiments, we recommend transferring between LLMs of similar scales or from larger to smaller-scale LLMs.
>
> > **4. Including Baselines in Human Evaluation**
>
> Thank you very much for the helpful suggestion! We will certainly integrate the NFP baselines into our demo to facilitate more comprehensive human evaluation.
>
> > **Q: Clarification on the End of Section 2**
>
> In Section 3.1, we defined that if an LLM-generated answer fails to convey the queried fact, it is a non-factual response. Therefore, the output statement “I apologize, but I don’t have information on ...” should be considered as non-factual. Our intent is not to criticize this statement but to clarify that they do not convey the queried fact. We will better claim this in the next version.
>
> [1] Alex Mallen, et al. When not to trust language models: Investigating effectiveness of parametric and non-parametric memories. In ACL, pp. 9802–9822, 2023.
>
> [2] Mert Yüksekgönül, et al. Attention satisfies: A constraint-satisfaction lens on factual errors of language models. In ICLR, 2024.
>
> [3] Saurav Kadavath, et al. Language models (mostly) know what they know. CoRR, abs/2207.05221, 2022.
>
> ---
>
> We sincerely thank you again for your detailed review and valuable suggestions. We would be happy to engage in further discussion and address any additional questions you may have.
>
> Sincerely,
>
> All authors

---

> > ### Comment · Reviewer_yYWm · 2024-11-26
> >
> > Thanks for the detailed response. I decide to keep my rating unchanged.

---

> > > ### Author Response · Authors · 2024-11-26
> > > **Request for Clarification on Remaining Concerns**
> > >
> > > Dear Reviewer yYWm,
> > >
> > > Thank you for your feedback. We noticed that you have decided to keep your rating unchanged. We would greatly appreciate it if you could clarify which concerns remain unresolved.
> > >
> > > Your insights are crucial for helping us improve our work, and we are more than willing to engage in further discussions with you.
> > >
> > > Looking forward to hearing from you.
> > >
> > > Best regards,
> > >
> > > All authors

---

> ### Author Response · Authors · 2024-11-25
> **Kind Request for Feedback**
>
> Dear Reviewer yYWm,
>
> Thank you very much for taking the time to review our paper. We sincerely appreciate your valuable comments and have provided detailed responses to address your concerns.
>
> We would greatly appreciate it if you could let us know whether our responses have addressed your concerns. If there are any remaining questions, we would be more than happy to provide further clarification and work towards resolving them.
>
> Thank you once again for your time, effort, and consideration, and we look forward to your valuable feedback.
>
> Best regards,
>
> All Authors

---

> ### Author Response · Authors · 2024-11-25
> **A Kind Reminder to Check Our Responses**
>
> Dear Reviewer yYWm,
>
> As the discussion period nearing its end, we notice that we have not yet received your feedback on our responses.
>
> We tried our best to address your concerns with detailed explanations. We kindly request you to check our responses and consider reassessing our paper.
>
> Thank you once again for your time and effort, and we look forward to hearing from you.
>
> Best regards,
>
> All authors

---

### Official Review · Reviewer_3fBW · 2024-11-06

**Soundness:** 2
**Presentation:** 3
**Contribution:** 2
**Rating:** 3
**Confidence:** 4

**Summary:**

The paper introduces a model method (FacLens) to predict the likelihood of language models (LMs) generating non-factual responses before generation occurs, a task called non-factuality prediction (NFP). This work claims that, unlike traditional non-factuality detection (NFD) methods that probe response representations, FacLens probes the question's hidden representations to make non-factuality predictions. FacLens can be adapted to different LMs by leveraging unsupervised domain adaptation techniques, which reduces the resource-intensive need to generate new labeled data for each model. The authors conduct experiments across four models and three datasets to demonstrate FacLens's superior performance and efficiency compared to baseline methods.

**Strengths:**

1. FacLens can be adapted to different LMs by leveraging unsupervised domain adaptation techniques, which reduces the resource-intensive need to generate new labeled data for each model.
2. The authors present a shift from traditional non-factuality detection (NFD) to non-factuality prediction (NFP). They show that models are internally aware of whether they can accurately answer a question before generation.

**Weaknesses:**

1. The authors claim that different LLMs share similar cognitive patterns in terms of knowledge awareness, as they rely on transformer-based architectures. However, not all LMs use the same architecture; for instance, recent MoE architectures, which have gained significant popularity, replace feed-forward networks with MoE modules. It is essential to study MoE-based models to examine if this claim holds. Additionally, the proof of this hypothesis is unclear and not convincing and needs further support.
2. Apart from the domain adaptation techniques, FacLens’s development heavily relies on previous work and lacks substantial novelty.
3. The overall performance gain compared to baselines, particularly SAPLMA, is marginal, and so there is no compelling evidence that probing question hidden representations leads to better non-factuality prediction.
4. In the main experiments (Table 1), NFD baselines are excluded, and only a selected set of methods categorized under NFP are reported.
5. The experiments do not represent a practical LM generation setting, as they are limited to a set of short-form QA datasets. While the authors define the NFP task, they compare it with naive baselines, such as entity popularity, and do not consider more sophisticated methods developed for factuality improvement using model internals.
6. Some findings, such as LLMs generally recognizing “whether they know” in their middle layers, have been previously reported and are not new findings.

Overall, this paper lacks significant contributions, and the limited experimental setup and marginal performance gains make it challenging to claim that the proposed method is more effective than its existing counterparts.

**Questions:**

Please refer to the weaknesses for clarification. Also, the paper has multiple typos that can be addressed.

---

> ### Author Response · Authors · 2024-11-18
> **Response to Reviewer 3fBW (1/2)**
>
> Dear reviewer 3fBW,
>
> We sincerely appreciate your effort in reviewing our paper. Here we will carefully provide responses to your concerns.
>
> > **1. Motivation Behind FacLens**
>
> ***- Why not use the MoE architecture:*** Non-MoE dense LLMs remain widely studied and applied. Following previous works (i.e., the baselines), we conducted experiments on non-MoE dense LLMs. Actually, there is no fundamental difference between MoE-based and dense LLMs, as both use the Transformer architecture, share training schemes (pre-training, SFT, RLHF), and overlap in training corpora. Thus, our work has the potential to be applied to MoE-based LLMs, though exploring this is beyond the scope of this paper and will be addressed in future work.
>
> ***- Proof of the hypothesis that LLMs share similar cognitions:*** The rationality of our work is based on the transferability of FacLens across different LLMs, instead of proving the hypothesis that LLMs share similar cognitions. This hypothesis only serves as a motivating inspiration. In terms of our focus, we have conducted comprehensive experiments in Section 4.2 to validate the transferability of FacLens.
>
> > **2. Novelty**
>
> Our work is the first to explore an **efficient and transferable** approach for ante-hoc non-factuality prediction (NFP). **Reviewer GNy3 explicitly acknowledged the novelty of our work**, emphasizing that this kind of transferability is importance in the NLP community. Unlike previous works, our work **simultaneously offers several key features**: 1) it is not limited to specific question templates, 2) it achieves promising results with a lightweight architecture, and 3) it supports unsupervised domain adaptation across LLMs for efficient development.
>
> > **3. Performance Improvement**
>
> Our work has **clear improvements** over existing works in practical applications **(efficiency beyond performance)** due to the following reasons.
>
> ***- Superiority of the ante-hoc (NFP) method FacLens over post-hoc (NFD) methods:*** In Figure 2, we compare the ante-hoc method (FacLens) with post-hoc methods (SAPLMA and INSIDE). Unlike post-hoc methods, which rely on costly answer generation, the ante-hoc method **avoids inference costs** and **controls risks in advance**. As shown in Figure 2, **despite post-hoc methods having more information (i.e., generated answers), FacLens still performs better.**
>
> ***- Improvement compared to NFP baselines:*** Table 1 shows that FacLens achieves clear performance gains over most baselines. While the performance gains over LoRA and Self-Evaluation are slightly smaller, FacLens significantly outperforms both baselines in terms of training **efficiency** (see Table 2), which is a crucial factor for practical application.
>
> > **4. Baselines**
>
> NFD models (post-hoc) are fundamentally different from NFP models (ante-hoc) in their settings (see the 3rd response). Therefore, comparing FacLens with NFD models in Table 1 is not necessary. To highlight the advantages and potential of the ante-hoc approach, we have compared FacLens with typical NFD methods in Figure 2.
>
> For the NFP baselines, we tried to systematically investigate the related work and compared FacLens with existing NFP baselines. We are open to include more works if you have any suggestions.
>
>
> > **5. Experimental Settings**
>
> ***- Form of answers:*** The factual QA scenario naturally favors short answers, as it focuses on querying specific real-world facts, such as names, numbers, birthplaces, dates, and so on. Additionally, most mainstream factual QA datasets adopt short answers. For these reasons, and to ensure a fair comparison, this paper follows our baselines to focus on common factual QA with typically short answers.
>
> While general QA with long-form answers (e.g., text summarization or literary creation) may include factual errors in generated responses, addressing factual errors in this context is another problem requiring a separate methodology, which presents a promising direction for future research.
>
> ***- NFP baselines:*** We have done our best to investigate related work and compared with existing NFP baselines. They are not all naive methods, **including advanced baselines** using model internals **such as SAT Probe [1] published on ICLR 2024**.
>
> [1] Mert Yüksekgönül, et al. Attention satisfies: A constraint-satisfaction lens on factual errors of language models. In ICLR, 2024.
>
>
> > **6. New Findings**
>
> While existing studies have shown that hidden representations in middle layers have good properties for other tasks, our work is the first to uncover that hidden question representations in middle layers contribute to the NFP task.
>
> Besides, this paper has several useful findings. For example, (1) hidden question representations from different LLMs share transferable NFP patterns (**approved by other reviewers**), and (2) ante-hoc method potentially perform better than the post-hoc method.
>
> > **Q1: Typos**
>
> We will proofread the paper again and fix the typos in the paper.

---

> ### Author Response · Authors · 2024-11-19
> **Response to Reviewer 3fBW (2/2)**
>
> We sincerely thank you again for your detailed review. We hope our responses will address your concerns. We are pleased to engage in further discussion and address any additional questions you may have.
>
> Sincerely,
>
> All authors

---

> ### Author Response · Authors · 2024-11-25
> **Kind Request for Feedback**
>
> Dear Reviewer 3fBW,
>
> Thank you very much for taking the time to review our paper. We sincerely appreciate your valuable comments and have provided detailed responses to address your concerns.
>
> We would greatly appreciate it if you could let us know whether our responses have addressed your concerns. If there are any remaining questions, we would be more than happy to provide further clarification and work towards resolving them.
>
> Thank you once again for your time, effort, and consideration, and we look forward to your valuable feedback.
>
> Best regards,
>
> All Authors

---

> ### Author Response · Authors · 2024-11-25
> **A Kind Reminder to Check Our Responses**
>
> Dear Reviewer 3fBW,
>
> As the discussion period nearing its end, we notice that we have not yet received your feedback on our responses.
>
> We tried our best to address your concerns with detailed explanations. We kindly request you to check our responses and consider reassessing our paper.
>
> Thank you once again for your time and effort, and we look forward to hearing from you.
>
> Best regards,
>
> All authors

---

### Author Response · Authors · 2024-12-04
**Summary of Key Advantages of This Work and Responses to Reviewers and Chairs**

Dear Reviewers and Chairs,

We sincerely appreciate the time and effort you have invested in reviewing our paper. We would like to summarize the key advantages of our work, which we hope will help you reassess it.

- **Ante-hoc Risk Control and Superior Performance**: While post-hoc methods leverage more information (i.e., the LLM-generated answers), FacLens achieves comparable and even better performance (see Figure 2). This demonstrates that FacLens can not only avoid inference costs but also proactively control risks associated with non-factual responses.

- **Pioneering Transferability in NFP (Acknowledged by All Reviewers)**: FacLens is the first work to effectively validate transferability in Non-Factuality Prediction (NFP), leveraging unsupervised domain adaptation to adapt to different LLMs without requiring new labeled data for each LLM. This innovative approach significantly improves efficiency.

- **Lightweight and Effective**: FacLens is a streamlined model that outperforms previous NFP models in both effectiveness and efficiency, making it  suitable for practical applications.

- **Well-Thought Experiments (Supported by Reviewer GNy3)**: The experiments are carefully designed and comprehensive, showcasing FacLens's effectiveness and efficiency across multiple benchmarks.

**We particularly appreciate Reviewer GNy3's acknowledgment that the domain adaptation of NFP models across LLMs is an important area of NLP research and should be encouraged across the community.**

In response to the reviewers' concerns, we have provided detailed clarifications and additional experiments. While we did not receive further detailed replies, we hope that the concerns have been adequately addressed. The instructive comments in the reviews can help us better improve this work. In response to Reviewer aR8Q’s comment, we conducted additional experiments on HotpotQA and TriviaQA, and the results continue to demonstrate the effectiveness of our approach.

Thank you once again for your efforts on reviewing this work. We hope our final response addresses your concerns effectively.

Best regards,

All authors

---

### Meta-Review · Area_Chair_sUdK · 2024-12-18

**Metareview:**

All reviewers agree that the paper provide a novel methodology to "predict" whether an LLM produce a nonfactual response based on the question's hidden representations. However, there are a few short comings that need to be addressed to make this work more complete. In particular,

- As provided by one of the reviewers, the scope of the work seem limited to short QAs. However, it is known that hallucination can be an issue for other tasks like summarization or when the answer is more involved than a single answer (which is more common use case of modern LLMs compared to classical Q&A systems). For example, what if the question ask to list all the grand slams won by a particular tennis player? How does this work is applicable there when the answer is almost correct except for a single item? If this work can be adopted to such use cases it becomes more practical.
- Related to the first item, how does this work is applicable in chat cases? How does the previous context can change hidden representations and maybe cause this method to have way too many false negatives?
- As mentioned by another reviewer, it is not clear how this transferability works. It would be good to check that after different fine-tuning/RLHF on the same pretrained model as well to see what affects that transferability. For example, if you take a model and train it on triviaQA test set, I assume the model should always produce the correct answer, would be good to see what such case tell about the transferability of that method compared to the same pretrained model finetuned without triviaqa test set.

**Additional Comments On Reviewer Discussion:**

None

---

### Decision · Program_Chairs · 2025-01-22

Reject